# Degradation-aware Dynamic Schrödinger Bridge for Unpaired Image Restoration

**Jingjun Yi**[1,2]*, **Qi Bi**[3✉], **Hao Zheng**[4✉], **Huimin Huang**[4], **Yixian Shen**[3], **Haolan Zhan**[5], **Wei Ji**[6],
**Yawen Huang**[1], **Yuexiang Li**[7], **Xian Wu**[4], **Yefeng Zheng**[1✉]

[1] Westlake University, China, [2]University of Alberta, Canada
[3]University of Amsterdam, the Netherland, [4]Tencent Jarvis Lab, China
[5]Monash University, Australia, [6]Yale University, the United States, [7]University of Macau, Macau
q.bi@ieee.org, howzheng@tencent.com, zhengyefeng@westlake.edu.cn

## Abstract

Image restoration is a fundamental task in computer vision and machine learning, which learns a mapping between the clear images and the degraded images under various conditions (e.g., blur, low-light, haze). Yet, most existing image restoration methods are highly restricted by the requirement of degraded and clear image pairs, which limits the generalization and feasibility to enormous real-world scenarios without paired images. To address this bottleneck, we propose a Degradation-aware Dynamic Schrödinger Bridge (DDSB) for unpaired image restoration. Its general idea is to learn a Schrödinger Bridge between clear and degraded image distribution, while at the same time emphasizing the physical degradation priors to reduce the accumulation of errors during the restoration process. A Degradation-aware Optimal Transport (DOT) learning scheme is accordingly devised. Training a degradation model to learn the inverse restoration process is particularly challenging, as it must be applicable across different stages of the iterative restoration process. A Dynamic Transport with Consistency (DTC) learning objective is further proposed to reduce the loss of image details in the early iterations and therefore refine the degradation model. Extensive experiments on multiple image degradation tasks show its state-of-the-art performance over the prior arts.

## 1 Introduction

Image restoration, encompassing tasks like denoising, deblurring, and dehazing, is one of the most crucial yet fundamental challenges in computer vision [8, 28, 54]. A traditional paradigm is supervised learning, where paired data of clear and degraded images are used to train models. However, acquiring large-scale paired datasets in many real-world scenarios is often impractical due to the vast diversity of degradation conditions and the absence of ground truth. Unpaired image restoration methods, which learn mappings between unpaired degraded and clear image sets, have gained significant attention in recent years as an alternative solution [30, 58, 47]. However, one of its key challenges lies in the absence of explicit correspondences between the input and output images.

The realm of generative models, exemplified by the recent diffusion models [38, 17, 39, 40, 35, 4], has achieved remarkable progress over the past few years. Compared with the prior arts (e.g., generative adversarial networks [14], variational autoencoders [23]), these recent generative models are more capable to generate diverse and high-quality samples [50], and have shown great success in image restoration [7]. Unfortunately, diffusion models are still subject to a specific prior distribution, e.g., Gaussian, which prevents its full potential to various degradation conditions in image restoration.

---

*Jingjun Yi is affiliated with University of Alberta. This research was conducted when he was a research intern at Westlake University.

39th Conference on Neural Information Processing Systems (NeurIPS 2025).

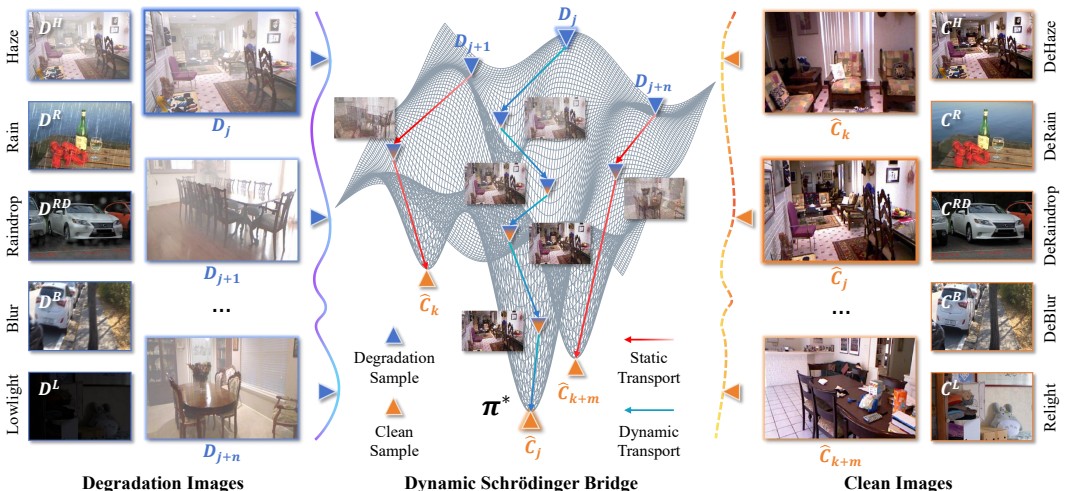

Figure 1: Dynamic Schrödinger bridge formulates image restoration as an optimal transport between the clear and degraded images, while maintaining the flexibility to handle unpaired data.

Schrödinger bridge (SB) provides a promising alternative by formulating the restoration problem as an optimal transport (OT) task between the clear image distribution and the degraded image distribution, while maintaining the flexibility to handle unpaired data [9, 6, 43]. In contrast to prior generative models, SB enables the translation between two arbitrary distributions without requiring paired training data [45, 29, 10, 41, 39, 5].

In light of these properties, the first research question naturally arises: *How to leverage SB for unpaired image restoration, while achieve as physically meaningful and realistic restoration as possible?* To address this challenge, we propose a Degradation-aware Optimal Transport (DOT) learning scheme, which builds on the entropy-regularized OT framework inherent in SB. The core objective of DOT is to reduce the accumulation of errors from unpaired SB during the iterative restoration process. By introducing a degradation model to learn the inverse process of restoration, DOT can amplify the degradations and artifacts produced by the restoration model, and impose an additional constraint based on this information.

Training the degradation model is particularly challenging due to the absence of paired training data, and this difficulty is compounded by the need for the model to generalize across different stages of the iterative restoration process. In the early stages of iteration, the predicted clean images often contain excessive artifacts, whereas in the later stages, the differences between consecutive steps become minimal. Both scenarios are suboptimal for effective degradation model training. To address these issues, we introduce Dynamic Transport with Consistency (DTC), which not only strengthens the constraints on the early stages, thereby reducing the loss of image details, but also refines the learning objective of the degradation model. This dual approach alleviates the challenges associated with learning the inverse restoration process and improves the overall effectiveness of the model.

Both components are incorporated as the proposed Degradation-aware Dynamic Schrödinger Bridge (DDSB). It ensures that the restoration process remains robust to variations in the degradation types, while preserving the integrity of the underlying image structure. The effectiveness of our method is validated through extensive experiments on multiple image restoration tasks, including denoising, deblurring, and dehazing, across a range of degradation conditions, in terms of both visual quality and quantitative metrics. In particular, our approach demonstrates superior generalization to unseen degradation types, providing a more robust solution for real-world image restoration tasks.

In a nutshell, our contributions can be summarized as follows.

- This paper opens up a new direction for applying Schrödinger Bridge in unpaired image restoration, providing a promising solution to the limitations of existing techniques.

- A Degradation-aware Dynamic Schrödinger Bridge (DDSB) is developed for unpaired image restoration, which incorporates a degradation model to reduce the error accumulation during the transport process.

- A Dynamic Transport with Consistency (DTC) learning objective is proposed to optimize the training of the degradation model.
- Extensive experiments on multiple image restoration benchmarks show its state-of-the-art performance.

## 2 Related Works

**Schrödinger Bridge** (SB) problem, also known as entropy-regularized optimal transport [37, 24], aims to learn a stochastic process between an initial distribution and a specified terminal distribution over time under the guidance of a reference measure. Owing to its property to allow arbitrary choices of initial and terminal distributions, SB has fostered the development of a variety of generative modeling problems, to name a few, iterative proportional fitting [9, 44], Riemannian manifolds [42], image translation [41], path sampling [55, 36], unpaired transport [22], under both supervised settings [15, 29, 10] and unsupervised settings [45]. In contrast to earlier work [49], the proposed method innovatively integrates dynamic multi-step transport modeling and trajectory consistency regularization. Besides, it is capable to handle various degradation types over deraining in [49].

**Unpaired Image-to-Image (I2I) Translation** aims to generate a target image that preserves the structural similarity of the source image without pair-wise input [59, 19]. Early works usually leverage geometric consistency [12] and mutual information regularization [3]. More recently, contrastive unpaired translation (CUT) and its variants [32, 21, 46, 57] have advanced I2I tasks by refining patch-wise regularization. Santa [51] introduces shortest-path constraints, while DN [18] applies dense normalization for translation. However, recent works usually conduct unpaired I2I under a resolution no higher than $128 \times 128$ pixels. In contrast, under our image restoration task, I2I requires substantially higher resolutions, which poses significant challenges. In addition, the proposed method is conceptually distinct from the previous circle consistency learning objective (e.g., CycleGAN [59]) by further imposing adaptive constraint on the SB path to align with plausible degradation outcomes and introducing temporal consistency across intermediate SB steps.

**Unpaired Image Restoration** aims to recover clean images from degraded ones without requiring aligned supervision [58]. Classical priors like DCP [16] rely on statistical assumptions, while recent domain-specific approaches, like YOLY [26] and USID-Net [27], exploit layer disentanglement and uncertainty-aware representations. CycleGAN [59] pioneered the use of cycle-consistency loss, while later works incorporated depth-aware degradation modeling [53, 56], mutal information maximization [32], and degradation feature decoupling [47]. However, the reliance on adversarial losses or handcrafted constraints makes it difficult to enforce degradation consistency throughout the restoration forward trajectory. In contrast, our method opens a new direction by introducing Schrödinger bridge to unpaired restoration.

## 3 Preliminaries

**Schrödinger Bridge.** Given two probability distributions $\pi_0$ and $\pi_1$ over $\mathbb{R}^d$, the Schrödinger Bridge problem seeks the most likely stochastic trajectory $\{x_t\}_{t \in [0,1]}$ that evolves from $\pi_0$ to $\pi_1$. Let $\Omega$ denote the space of continuous paths in $\mathbb{R}^d$, and let $\mathcal{P}(\Omega)$ represent the collection of path measures. The SB problem can be formulated as the following entropy-regularized variational problem:

$$\mathbb{Q}^\star = \underset{\mathbb{Q} \in \mathcal{P}(\Omega)}{\arg\min} \, \mathrm{KL}(\mathbb{Q} \, \| \, \mathbb{W}^\tau) \quad \text{s.t.} \quad \mathbb{Q}_0 = \pi_0, \quad \mathbb{Q}_1 = \pi_1, \tag{1}$$

where $\mathbb{W}^\tau$ is the Wiener measure with diffusion parameter $\tau$, and $\mathbb{Q}_t$ represents the marginal at time $t$. The solution $\mathbb{Q}^\star$ defines the Schrödinger Bridge connecting $\pi_0$ to $\pi_1$.

Two perspectives—stochastic control and static coupling—offer foundational insight into the SB problem and guide our model design. Both are central to understanding the shortcomings of prior SB methods and motivating our approach.

**Stochastic Control Formulation.** From the viewpoint of stochastic control [33], the SB process $\{x_t\} \sim \mathbb{Q}^\star$ satisfies a stochastic differential equation $dx_t = u_t^\star dt + \sqrt{\tau} dw_t$, where $u_t^\star$ is the optimal control minimizing the expected energy of the drift:

$$u_t^\star = \underset{u}{\arg\min} \, \mathbb{E}\left[\int_0^1 \frac{1}{2}\|u_t\|^2 dt\right] \quad \text{s.t.} \quad \begin{cases} dx_t = u_t dt + \sqrt{\tau} dw_t \\ x_d \sim \pi_0, \quad x_c \sim \pi_1. \end{cases} \tag{2}$$

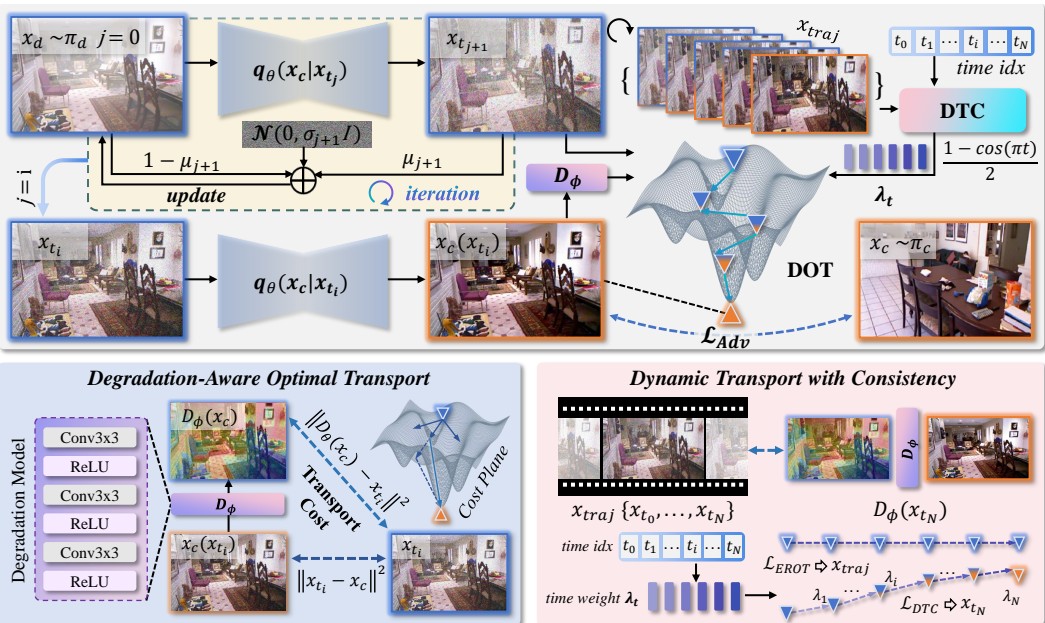

Figure 2: Framework overview of the proposed Degradation-aware Dynamic Schrödinger Bridge (DDSB). On top of the unpaired Schrödinger Bridge model, it presents two novel components, namely, Degradation-aware Optimal Transport (DOT) and Dynamic Transport with Consistency (DTC).

This formulation reveals that SB paths are minimum-action stochastic processes constrained to match boundary distributions. The resulting trajectory is both Markovian and converges to the deterministic optimal transport flow as $\tau \to 0$, with $\tau$ determining the stochasticity level.

**Static Coupling.** SB also admits a static formulation based on endpoint marginals. If the joint distribution at times $t = 0$ and $t = 1$ is known, denoted $\mathbb{Q}_{01}^{\star}$, then intermediate states are conditionally Gaussian, given by $p(\boldsymbol{x}_t \mid \boldsymbol{x}_d, \boldsymbol{x}_c) = \mathcal{N}(\boldsymbol{x}_t \mid (1-t)\boldsymbol{x}_d + t\boldsymbol{x}_c, \tau t(1-t)\mathbf{I})$. This allows trajectory simulation by sampling from the coupling $\mathbb{Q}_{01}^{\star}$, and the marginal density of $\boldsymbol{x}_t$ given $\boldsymbol{x}_d$ is thus:

$$p(\boldsymbol{x}_t \mid \boldsymbol{x}_d) = \int p(\boldsymbol{x}_t \mid \boldsymbol{x}_d, \boldsymbol{x}_c) \, d\mathbb{Q}_{1|0}^{\star}(\boldsymbol{x}_c \mid \boldsymbol{x}_d). \tag{3}$$

The optimal coupling $\mathbb{Q}_{01}^{\star}$ satisfies the entropy-regularized optimal transport (EROT) formulation:

$$\mathbb{Q}_{01}^{\star} = \underset{\gamma \in \Pi(\pi_0, \pi_1)}{\arg\min} \, \mathbb{E}_{(\boldsymbol{x}_d, \boldsymbol{x}_c) \sim \gamma} \left[ \|\boldsymbol{x}_d - \boldsymbol{x}_c\|^2 \right] - 2\tau \mathcal{H}(\gamma), \tag{4}$$

where $\Pi(\pi_0, \pi_1)$ denotes the set of couplings with marginals $\pi_0$ and $\pi_1$, and $\mathcal{H}(\gamma)$ is the entropy.

## 4 Method

In unpaired image restoration, the source and target distributions represent degraded and clean visual domains, respectively. While Schrödinger Bridge (SB) offers a principled probabilistic framework to interpolate between two distributions via entropy-regularized optimal transport (EROT), conventional formulations often neglect physical degradation priors that are crucial for realism in restoration tasks. We propose a novel formulation *Degradation-aware Dynamic Schrödinger Bridge (DDSB)* that explicitly integrates a differentiable degradation model within the SB formulation. DDSB enforces that the restored outputs remain consistent with their degraded inputs under learned degradation dynamics. The framework overview is shown in Fig. 2.

### 4.1 Unpaired Schrödinger Bridge (UNSB)

The DDSB framework formulates the Schrödinger Bridge as a composition of learnable generative transitions that iteratively transport samples from a degraded distribution toward the clean image

distribution. Let $\boldsymbol{x}_d \sim \pi_d$ denote a real degraded image, where $\pi_d$ is the degraded image distribution. The objective is to recover its corresponding clean counterpart $\boldsymbol{x}_c \in \pi_c$, with $\pi_c$ representing the clean image distribution. Consider a discretized time partition $\{t_i\}_{i=0}^N$ of the unit interval $[0,1]$, where $t_0 = 0$, $t_N = 1$, and $t_i < t_{i+1}$. The UNSB process can be simulated via a Markov chain:

$$p(\{\boldsymbol{x}_{t_n}\}) = p(\boldsymbol{x}_{t_N}|\boldsymbol{x}_{t_{N-1}}) \cdots p(\boldsymbol{x}_{t_1}|\boldsymbol{x}_{t_0})p(\boldsymbol{x}_{t_0}). \tag{5}$$

Here, $\boldsymbol{x}_{t_0}$ corresponds to the degraded input $\boldsymbol{x}_d$, while $\boldsymbol{x}_{t_N}$ approximates the clean image $\boldsymbol{x}_c$. To approximate $p(\boldsymbol{x}_{t_i})$, a conditional generator $q_{\theta_i}(\boldsymbol{x}_c|\boldsymbol{x}_{t_i})$ is introduced, where $\theta_i$ denotes the network parameters at step $i$. This defines a joint distribution between the latent state and the clean target as:

$$q_{\theta_i}(\boldsymbol{x}_{t_i}, \boldsymbol{x}_c) = q_{\theta_i}(\boldsymbol{x}_c|\boldsymbol{x}_{t_i})p(\boldsymbol{x}_{t_i}), \quad q_{\theta_i}(\boldsymbol{x}_c) = \mathbb{E}_{p(\boldsymbol{x}_{t_i})}[q_{\theta_i}(\boldsymbol{x}_c|\boldsymbol{x}_{t_i})]. \tag{6}$$

Given a sample $\boldsymbol{x}_{t_j} \sim q_\theta(\boldsymbol{x}_{t_j})$, the generator predicts a clean estimate $\boldsymbol{x}_c(\boldsymbol{x}_{t_j}) \sim q_\theta(\boldsymbol{x}_c|\boldsymbol{x}_{t_j})$. A new sample $\boldsymbol{x}_{t_{j+1}}$ is then obtained by interpolating $\boldsymbol{x}_{t_j}$ with $\boldsymbol{x}_c(\boldsymbol{x}_{t_j})$ and injecting noise:

$$p(\boldsymbol{x}_{t_{j+1}}|\boldsymbol{x}_c, \boldsymbol{x}_{t_j}) = \mathcal{N}\left(\boldsymbol{x}_{t_{j+1}} \mid \mu_{j+1}\boldsymbol{x}_c + (1-\mu_{j+1})\boldsymbol{x}_{t_j}, \sigma_{j+1}^2 \boldsymbol{I}\right), \tag{7}$$

where $\mu_{j+1} = \frac{t_{j+1}-t_j}{1-t_j}$ controls the interpolation weight, and the noise scale is computed as $\sigma_{j+1}^2 = \mu_{j+1}(1 - \mu_{j+1})\tau(1 - t_j)$, with $\tau$ being a hyperparameter.

By iteratively applying this transition for $j = 0, \ldots, i - 1$, sample $\boldsymbol{x}_{t_i}$ is generated. When the generator $q_{\theta_i}$ is sufficiently optimized, its marginal $q_{\theta_i}(\boldsymbol{x}_{t_i})$ closely approximates the target $p(\boldsymbol{x}_{t_i})$. The sequence $\{\boldsymbol{x}_c(\boldsymbol{x}_{t_i})\}_{i=0}^{N-1}$ thus forms a progressively refined trajectory toward clean image reconstruction. The optimal parameters $\theta$ are obtained by optimizing the Schrödinger Bridge via entropy-regularized optimal transport (EROT), defined as

$$\mathcal{L}_{\text{EROT}} = \mathbb{E}_{\boldsymbol{x}_c, \boldsymbol{x}_d \sim \pi}\left[\|\boldsymbol{x}_d - \boldsymbol{x}_c\|^2\right] - 2\tau\,\mathcal{H}(\pi) + \text{KL}(q_\theta(\boldsymbol{x}_c)\,\|\,p(\boldsymbol{x}_c)), \pi \sim \Pi(\pi_d, \pi_c). \tag{8}$$

Specifically, the first term of $\mathcal{L}_{\text{EROT}}$ seeks a stochastic coupling $\pi^\star$ between $\pi_d$ and $\pi_c$:

$$\pi^\star = \underset{\pi \in \Pi(\pi_d, \pi_c)}{\arg\min} \mathbb{E}_{(\boldsymbol{x}_d, \boldsymbol{x}_c) \sim \pi}\left[\|\boldsymbol{x}_d - \boldsymbol{x}_c\|^2\right] - 2\tau\,\mathcal{H}(\pi), \tag{9}$$

where $\mathcal{H}(\pi)$ is the joint entropy and $\tau$ is a temperature hyperparameter. It measures the reconstruction error between the predicted clean sample and the clean targe.

To ensure that the clean images inferred from $q_\theta(\boldsymbol{x}_c|\boldsymbol{x}_{t_i})$ match the target distribution $p(\boldsymbol{x}_c)$, the second term uses an adversarial Kullback-Leibler (KL) regularization, given by:

$$\mathcal{L}_{\text{adv}} = \text{KL}(q_\theta(\boldsymbol{x}_c)\,\|\,p(\boldsymbol{x}_c)), \tag{10}$$

which enforces marginal alignment at the clean endpoint. This loss acts as a global guidance signal complementing local transport fidelity, encouraging the generator to produce clean estimates that are statistically indistinguishable from real data. In practice, this term is implemented via a discriminator network and optimized using a reverse KL objective.

## 4.2 Degradation-aware Optimal Transport

To enhance the Schrödinger Bridge optimization in unpaired restoration, we introduce *Degradation-aware Optimal Transport* (DOT), which integrates degradation consistency into the EROT framework. An important issue with the aforementioned SB process is the accumulation of image detail loss resulting from model inference. In particular, during the initial stages, predicting a clear image from a severely degraded input often introduces new blur, over-smoothing, or unnatural textures. These artifacts are propagated to subsequent iterations via Eq. 7, thereby affecting the final restoration.

Let $\pi_d$ and $\pi_c$ denote the marginal distributions over degraded and clean images, respectively. The classical static Schrödinger Bridge seeks a coupling $\pi$ minimizing an entropy-regularized cost as defined in Eq. 9. We enhance this cost by introducing a degradation-alignment term, yielding the DOT energy:

$$c_{\text{DOT}}(\boldsymbol{x}_d, \boldsymbol{x}_c) := \|\boldsymbol{x}_d - \boldsymbol{x}_c\|^2 + \lambda \cdot \|D_\phi(\boldsymbol{x}_c) - \boldsymbol{x}_d\|^2, \tag{11}$$

where $D_\phi$ is a learnable degradation model, and $\lambda$ weights degradation fidelity. Model $D_\phi$ is used to amplify the existing degradations in the image, that is, to learn the inverse process of image restoration. On the one hand, compared to artifacts introduced by model predictions, certain natural

degradation processes, such as haze and rain, are easier to learn. On the other hand, for new types of degradation generated during prediction, amplification makes their differences from the original $\boldsymbol{x}_d$ more pronounced. Through this component, we impose constraints on the degradations and artifacts produced by the model. Plugging Eq. 11 into Eq. 9, we obtain:

$$\pi^\star = \underset{\pi \in \Pi(\pi_d, \pi_c)}{\arg\min} \; \mathbb{E}_{(\boldsymbol{x}_d, \boldsymbol{x}_c) \sim \pi} \left[ c_{\text{DOT}}(\boldsymbol{x}_d, \boldsymbol{x}_c) \right] - 2\tau\, \mathcal{H}(\pi). \tag{12}$$

To align this with UNSB's iterative structure, we implement DOT at the selected transport step $t_i$, where $\boldsymbol{x}_{t_i}$ is the intermediate state sampled from the evolving distribution. Let $q_\theta(\boldsymbol{x}_c | \boldsymbol{x}_{t_i})$ denote the distribution of generated clean image. We define the DOT loss at step $t_i$ as:

$$\mathcal{L}_{\text{DOT}}(\theta, t_i) := \mathbb{E}_{q_\theta(\boldsymbol{x}_{t_i}, \boldsymbol{x}_c) \sim \pi} \left[ \|\boldsymbol{x}_{t_i} - \boldsymbol{x}_c(\boldsymbol{x}_{t_i})\|^2 + \lambda \cdot \|\boldsymbol{x}_{t_i} - D_\phi(\boldsymbol{x}_c(\boldsymbol{x}_{t_i}))\|^2 \right]$$
$$- 2\tau(1 - t_i) \cdot \mathcal{H}(q_\theta(\boldsymbol{x}_{t_i}, \boldsymbol{x}_c(\boldsymbol{x}_{t_i})) + \text{KL}(q_\theta(\boldsymbol{x}_c | \boldsymbol{x}_{t_i}) \| p(\boldsymbol{x}_c)). \tag{13}$$

### 4.3 Dynamic Transport with Consistency

DOT imposes the $\mathcal{L}_{\text{DOT}}$ constraint at a randomly sampled time point $t_i$. However, during the iterative image restoration process, the initial stages are significantly more challenging than the later iterations. Artifacts introduced by the restoration model are also more likely to occur in these early stages. Therefore, it is necessary to place greater emphasis on the initial stages during the training of the restoration model. To this end, we further propose *Dynamic Transport with Consistency* (DTC), which provides supervision at every step of the restoration trajectory.

Specifically, for a sampled time point $t_{N_s}$, we denote the restoration trajectory as $\boldsymbol{x}_{\text{traj}} = \{\boldsymbol{x}_{t_0}, \boldsymbol{x}_{t_1}, \ldots, \boldsymbol{x}_{t_{N_s}}\}$. We apply $\mathcal{L}_{\text{DOT}}$ at each step of the trajectory, while the first item in $\mathcal{L}_{\text{DOT}}$ is replaced by $\mathcal{L}_{\text{DTC}}$, which is defined as:

$$\mathcal{L}_{\text{DTC}} = \|\boldsymbol{x}_{t_i} - \boldsymbol{x}_c(\boldsymbol{x}_{t_i})\|^2 + \lambda_{t_i} \cdot \|\boldsymbol{x}_{t_i} - D_\phi(\boldsymbol{x}_{t_{N_s}})\|^2. \tag{14}$$

Here $\lambda_{t_i}$ is the weight for the degradation-alignment term, which is computed by

$$\lambda_t = \lambda \cdot \frac{1 - \cos(\pi t)}{2}, \quad t \in [0, 1]. \tag{15}$$

Compared with $\mathcal{L}_{\text{DOT}}$, $\mathcal{L}_{\text{DTC}}$ replaces $\lambda \|\boldsymbol{x}_{t_i} - D_\phi(\boldsymbol{x}_c(x_{t_i}))\|^2$ with $\lambda_{t_i} \|\boldsymbol{x}_{t_i} - D_\phi(\boldsymbol{x}_{t_{N_s}})\|^2$. The learning objective of $D_\phi(\boldsymbol{x}_{t_{N_s}})$ becomes $\sum_{i=0}^{N_s-1} \lambda_{t_i} \boldsymbol{x}_{t_i} / \sum_{i=0}^{N_s-1} \lambda_{t_i}$. There are two main reasons behind this modification. First, the $\boldsymbol{x}_c(x_{t_i})$ in initial stage may contain unnatural scenes, and such representations are not suitable for training the degradation model $D_\phi$, whereas the end point of the trajectory $\boldsymbol{x}_{t_{N_s}}$ is of higher quality. In addition, for some steps, the difference between $\boldsymbol{x}_{t_i}$ and $\boldsymbol{x}_{t_{i+1}}$ is not significant, which also makes them less suitable for training $D_\phi$. Therefore, we use a weighted average over the entire trajectory as the learning objective, where time steps closer to $t_{N_s}$ are assigned higher weights. This approach can, to some extent, reduce the difficulty of degradation learning.

**Understanding DTC.** In DTC, we enhance the focus on the early stages of the iterative process by supervising the entire trajectory. More importantly, we employ a constraint based on $D_\phi(\boldsymbol{x}_{t_{N_s}})$ to mitigate the loss of image details during restoration. Compared to using $\boldsymbol{x}_{t_0}$, $D_\phi(\boldsymbol{x}_{t_{N_s}})$ not only incorporates certain restorations upon $\boldsymbol{x}_{t_0}$, but also amplifies the degradations and artifacts introduced by the restoration model, making it a more effective constraint. In other words, the transport trajectory is supervised by penalizing the differences between adjacent SB states (i.e., pairwise consistency between $\boldsymbol{x}_{t_i}$ and $\boldsymbol{x}_{t_{i-1}}$). Compared to baseline UNSB, which only supervises the endpoints, our method densely regularizes the entire trajectory, especially the early steps, where errors are more likely to propagate. Therefore, even though the overall loss includes unweighted endpoint terms, the trajectory-level consistency term explicitly emphasizes earlier stages due to its design, providing stronger guidance when the model is most prone to drift.

### 4.4 Implementation Details

**Training Strategy.** We randomly select a time step $t_{N_s} \in \{t_0, \ldots, t_N\}$ and simulate the forward transport from the degraded sample $\boldsymbol{x}_d \sim \pi_0$ to obtain $\{\boldsymbol{x}_{t_0}, \boldsymbol{x}_{t_1}, \ldots, \boldsymbol{x}_{t_{N_s}}\}$. In each intermediate step, $\boldsymbol{x}_{t_i}$ is fed into the conditional generator $q_\theta(\boldsymbol{x}_c | \boldsymbol{x}_{t_i})$ so as to output a clean estimate $\boldsymbol{x}_c(\boldsymbol{x}_{t_i})$.

Together with a reference sample $\boldsymbol{x}_c \sim \pi_1$, the tuples $(\boldsymbol{x}_{t_i}, \boldsymbol{x}_c(\boldsymbol{x}_{t_i}))$ and $(\boldsymbol{x}_c, \boldsymbol{x}_c(\boldsymbol{x}_{t_i}))$ are used to compute $\mathcal{L}_{\mathrm{DOT}}(\theta, t_i)$ and $\mathcal{L}_{\mathrm{DTC}}(\theta, t_i)$. The entropy term is computed by the mutual information neural estimation (MINE) [2]. The KL divergence is implemented via adversarial learning, where $\boldsymbol{x}_c$ and $\boldsymbol{x}_c(\boldsymbol{x}_{t_i})$ serve as real and fake inputs, respectively, to a Markovian discriminator.

**Network Architecture.** The conditional generator $q_\theta$ is implemented using a U-Net architecture with instance normalization, where the encoder and decoder consists of five convolutional and deconvolutional layers, respectively. The input and sampled noise are concatenated as the input channels at the first layer, enabling conditioning on both the degraded input and latent noise during the restoration process. The degradation model is a lightweight convolutional neural network (CNN), which consists of three 64-channel convolutional layers with ReLU activation functions. Notice that, the degradation model is designed as a degradation guidance module rather than a reconstructor. It guides the restoration process by penalizing implausible states through constraints on the SB trajectory, making it highly effective for its purpose despite being small.

**Hyperparameters.** The training terminates after 400 epochs, with a batch size of 1. The initial learning rate is $2 \times 10^{-4}$ and decays to zero linearly. All the inputted images are firstly resized to $512 \times 512$ pixels and then the image intensity is normalized to $[-1, 1]$. The time interval $[0, 1]$ is discretized into $N=5$ uniform steps. The temperature parameter is fixed as $\tau=0.01$, and the balanced constant hyper-parameter of $\mathcal{L}_{\mathrm{DOT}}$ $\lambda$ is set to 0.01. For DTC, the weight of degradation term adopts a cosine annealing schedule. The generator adopts AdaIN layers and sinusoidal timestep embeddings, following DDGAN [50].

## 5 Experiments

### 5.1 Results on Multi-Task Image Restoration

**Datasets & Evaluation Protocols.** We adopt four restoration tasks: deraining, raindrop removal, deblurring, and low-light enhancement. Rain200L [52] is used for deraining. It includes 1,800 synthetic rainy images for training and 200 for testing. Raindrop [34] is used for raindrop removal. It consists of 1,119 paired images with and without raindrops on glass surfaces. The GoPro dataset [31] is widely used for image deblurring, containing 3,214 high-resolution blurred images ($1280 \times 720$ pixels). It is split into 2,103 samples for training and 1,111 for testing. LOL [48] is used for low-light enhancement. It consists of 500 image pairs captured under normal and low-light conditions. Following prior works [20], we use 485 pairs for training and 15 for testing. All images are resized to $512 \times 512$ pixels and the intensity is normalized before training. These datasets cover a broad spectrum of degradations and provide a comprehensive testbed for multi-task restoration. Following prior unpaired methods [53, 47], during training, for each degraded input image, we randomly sample a clean image from the set, excluding its ground-truth counterpart. The resulting image tuple is thus unaligned and used solely for learning the transport between unpaired distributions. Structural Similarity Index (SSIM) and Peak Signal-to-Noise Ratio (PSNR, in dB), are used for evaluation.

**Baselines.** We compare DDSB with a wide range of state-of-the-art unpaired image restoration methods. These include the classical prior-based approach DCP [16], and various learning-based methods such as CycleGAN [59], YOLY [26], USID-Net [27], RefineDNet [56], D$^4$ [53], CUT [32], Santa [51], ODCR [47], and DN [18]. We also include UNSB [22], which formulates cross-domain generation as a Schrödinger Bridge problem. Although Santa, DN, and UNSB were originally proposed for unpaired image translation, we adapt them to image restoration for a boarder comparison. For each task, all the methods are fine-tuned under the same configuration.

**Quantitative Evaluation.** Table 1 shows that DDSB achieves the best performance across all four unpaired image restoration tasks. DDSB consistently outperforms all competing unpaired methods in both PSNR and SSIM. Compared with the strongest baseline DN, DDSB achieves significant PSNR gains of +0.69 dB on deraining, +1.12 dB on de-raindrop removal, +1.80 dB on low-light enhancement, and +2.02 dB on deblurring. Corresponding SSIM improvements are +0.025, +0.007, +0.021, and +0.008, respectively. These consistent improvements, especially under challenging conditions like low-light and motion blur, validate the effectiveness of our dynamic Schrödinger Bridge framework with degradation consistency. Furthermore, compared with CUT and D$^4$, DDSB demonstrates superior robustness due to its explicit modeling of intermediate transport dynamics and degradation-aligned consistency.

Table 1: Quantitative comparison of DDSB with the state-of-the-art unpaired image restoration methods on multi-task restoration. Top three results are highlighted as `best` , `second` and `third` .

| Method | Derain [52] | | Deraindrop [34] | | Lowlight [48] | | Deblur [31] | |
|---|---|---|---|---|---|---|---|---|
| | PSNR (dB) | SSIM | PSNR (dB) | SSIM | PSNR (dB) | SSIM | PSNR (dB) | SSIM |
| DCP [16] | 13.25 | 0.705 | 18.92 | 0.752 | 15.93 | 0.743 | 12.97 | 0.702 |
| CycleGAN [59] | 21.28 | 0.796 | 20.55 | 0.787 | 14.03 | 0.781 | 19.10 | 0.735 |
| YOLY [26] | 15.72 | 0.714 | 14.71 | 0.748 | 13.16 | 0.762 | 16.28 | 0.717 |
| USID-Net [27] | 21.50 | 0.784 | 19.81 | 0.771 | 17.91 | 0.769 | 20.72 | 0.726 |
| RefineDNet [56] | 24.41 | 0.840 | 21.65 | 0.783 | 19.75 | 0.793 | 21.03 | 0.747 |
| D$^4$ [53] | 24.75 | 0.832 | 23.84 | 0.805 | 21.32 | 0.826 | 21.59 | 0.782 |
| CUT [32] | 24.22 | 0.815 | 23.51 | 0.827 | 22.90 | 0.804 | 21.26 | 0.766 |
| Santa [51] | 24.55 | 0.828 | 23.65 | 0.797 | 21.93 | 0.838 | 21.80 | 0.778 |
| UNSB [22] | 24.68 | 0.837 | 24.52 | 0.812 | 22.75 | 0.822 | 22.11 | 0.785 |
| ODCR [47] | 24.89 | 0.848 | 24.08 | 0.818 | 23.42 | 0.832 | 22.73 | 0.791 |
| DN [18] | 24.72 | 0.845 | 24.63 | 0.824 | 23.58 | 0.844 | 23.20 | 0.796 |
| **DDSB (ours)** | **25.41** | **0.870** | **25.75** | **0.831** | **25.38** | **0.865** | **25.22** | **0.804** |

Table 2: Non-parametric perceptual metric comparison of DDSB with the state-of-the-art unpaired image restoration methods. Top three results are highlighted as `best` , `second` and `third` .

| Method | Derain [52] | | Deraindrop [34] | | Lowlight [48] | | Deblur [31] | |
|---|---|---|---|---|---|---|---|---|
| | LPIPS | NIQE | LPIPS | NIQE | LPIPS | NIQE | LPIPS | NIQE |
| DCP [16] | 0.229 | 5.37 | 0.204 | 5.61 | 0.218 | 5.77 | 0.231 | 5.94 |
| CycleGAN [59] | 0.146 | 4.82 | 0.167 | 4.69 | 0.197 | 5.44 | 0.172 | 4.98 |
| YOLY [26] | 0.198 | 5.03 | 0.202 | 5.19 | 0.210 | 5.66 | 0.211 | 5.47 |
| USID-Net [27] | 0.152 | 4.75 | 0.172 | 4.92 | 0.188 | 5.31 | 0.162 | 4.88 |
| RefineDNet [56] | 0.104 | 4.36 | 0.147 | 4.68 | 0.179 | 5.22 | 0.150 | 4.71 |
| D$^4$ [53] | 0.098 | 4.28 | 0.124 | 4.50 | 0.155 | 5.04 | 0.139 | 4.61 |
| CUT [32] | 0.111 | 4.42 | 0.118 | 4.55 | 0.142 | 5.17 | 0.143 | 4.66 |
| Santa [51] | 0.096 | 4.33 | 0.109 | 4.39 | 0.138 | 5.09 | 0.130 | 4.54 |
| UNSB [22] | 0.085 | 4.18 | 0.096 | 4.27 | 0.145 | 5.02 | 0.126 | 4.49 |
| ODCR [47] | 0.076 | 4.09 | 0.091 | 4.21 | 0.131 | 4.88 | 0.121 | 4.43 |
| DN [18] | 0.079 | 4.11 | 0.082 | 4.17 | 0.124 | 4.85 | 0.118 | 4.40 |
| **DDSB (ours)** | **0.063** | **3.94** | **0.069** | **4.03** | **0.129** | **4.79** | **0.108** | **4.31** |

Table 3: Quantitative comparison of DDSB with the state-of-the-art unpaired dehazing methods on the generalized dehazing task, trained on SOTS-indoor, and the test result are shown. Cells where results are not available are replaced by "-". The time is measured on images of the size of $512 \times 512$ pixels using a single GPU.

| Method | SOTS-indoor [25] | | SOTS-outdoor [25] | | NH-HAZE 2 [1] | | Overhead | |
|---|---|---|---|---|---|---|---|---|
| | PSNR (dB) | SSIM | PSNR (dB) | SSIM | PSNR (dB) | SSIM | Para. (M) | Time (ms) |
| DCP [16] | 13.10 | 0.699 | 19.13 | 0.815 | 14.90 | 0.668 | - | - |
| CycleGAN [59] | 21.34 | 0.898 | 20.55 | 0.856 | 13.95 | 0.689 | 11.38 | 10.22 |
| YOLY [26] | 15.84 | 0.819 | 14.75 | 0.857 | 13.38 | 0.595 | 32.00 | - |
| USID-Net [27] | 21.41 | 0.894 | 23.89 | 0.919 | 15.62 | 0.740 | 3.780 | 31.01 |
| RefineDNet [56] | 24.36 | 0.939 | 19.84 | 0.853 | 14.20 | 0.754 | 65.80 | 248.5 |
| D$^4$ [53] | 25.42 | 0.932 | 25.83 | 0.956 | 14.52 | 0.709 | 10.70 | 28.08 |
| CUT [32] | 24.30 | 0.911 | 23.67 | 0.904 | 15.92 | 0.758 | 11.38 | 10.06 |
| Santa [51] | 25.01 | 0.923 | 24.21 | 0.945 | 16.02 | 0.749 | 11.43 | 136 |
| UNSB [22] | 25.68 | 0.930 | 25.30 | 0.954 | 16.10 | 0.753 | 14.42 | 0.212 |
| ODCR [47] | 26.32 | 0.945 | 26.16 | 0.960 | 17.56 | 0.766 | 11.38 | 10.14 |
| DN [18] | 26.25 | 0.947 | 26.18 | 0.962 | 17.15 | 0.769 | 11.40 | 87.7 |
| **DDSB (ours)** | **27.85** | **0.956** | **27.67** | **0.971** | **17.92** | **0.783** | 14.68 | 0.019 |

**Non-parametric Perceptual Evaluation.** We additionally report two non-parametric metrics that provide a more holistic assessment without relying on predefined statistical models. The Learned Perceptual Image Patch Similarity (LPIPS) measures perceptual distance in deep feature space using a pretrained network and requires ground-truth reference images. The Naturalness Image Quality Evaluator (NIQE) operates without any reference image by modeling natural image statistics. As shown in Table 2, we evaluate these two non-parametric metrics on our multi-task restoration benchmark. DDSB consistently outperforms prior methods, demonstrating its strong generalization and perceptual fidelity without relying on pixel-wise supervision.

**Qualitative Evaluation.** Visual comparisons on four unpaired restoration tasks are shown in Fig. 3. D$^4$ struggles to preserve structural details. UNSB yields inconsistent restorations. DN improves local textures, yet residual degradations remain, especially under blur and low-light conditions. ODCR

Table 4: Ablation study on each component. Evaluation metrics include PSNR↑ and SSIM↑.

| Component | | | SOTS-indoor | | SOTS-outdoor | |
|---|---|---|---|---|---|---|
| EROT | DOT | DTC | PSNR | SSIM | PSNR | SSIM |
| ✓ | ✗ | ✗ | 25.68 | 0.930 | 25.30 | 0.954 |
| ✗ | ✓ | ✗ | 27.23 | 0.953 | 27.04 | 0.963 |
| ✗ | ✗ | ✓ | 26.11 | 0.947 | 25.92 | 0.957 |
| ✗ | ✓ | ✓ | **27.85** | **0.956** | **27.67** | **0.971** |

Table 5: Impact of time step $N$.

| $N$ | SOTS-indoor | |
|---|---|---|
| | PSNR | SSIM |
| 2 | 26.18 | 0.943 |
| 3 | 26.91 | 0.950 |
| 4 | 27.42 | **0.957** |
| 5 | **27.85** | **0.956** |
| 6 | 27.09 | 0.949 |

Table 6: Impact of hyper-parameter $\lambda$.

| $\lambda$ | SOTS-indoor | |
|---|---|---|
| | PSNR | SSIM |
| 0.0001 | 25.76 | 0.940 |
| 0.001 | 27.12 | 0.950 |
| 0.01 | **27.85** | **0.956** |
| 0.1 | 27.24 | 0.949 |
| 1 | 26.91 | 0.946 |

Table 7: Impact of layer depth in $D_\phi$.

| $\lambda$ | SOTS-indoor | |
|---|---|---|
| | PSNR | SSIM |
| 2 | 25.37 | 0.926 |
| 3 | **27.85** | 0.956 |
| 5 | 27.69 | **0.958** |
| 7 | 26.70 | 0.933 |

Figure 3: Comparison of state-of-the-art unpaired methods on multi-task image restoration.

still falters under severe corruption such as dense raindrops or extreme darkness. In contrast, DDSB delivers perceptually sharper and more faithful restorations across all scenarios.

## 5.2 Results on Generalized Haze Removal

**Datasets and Evaluation Protocols.** To evaluate the generalization capability of DDSB under diverse degradation scenarios, we follow the setting of [47] by training on the Indoor Training Set (ITS) from RESIDE [25], and evaluating on the SOTS-outdoor (OTS) test set from NH-HAZE 2 [1], which jointly covers synthetic, artificial, and real-world domains. Specifically, RESIDE provides 13,990 synthetic hazy-clear image pairs in ITS and 500 outdoor test pairs in OTS. NH-HAZE 2 contains 25 image pairs with non-homogeneous haze for more challenging and realistic evaluation.

**Baselines.** We compare DDSB against a broad set of state-of-the-art image dehazing methods. In the unpaired setting, we include both traditional priors and recent deep models, including DCP [16], CycleGAN [59], CycleDehaze [11], YOLY [26], USID-Net [27], RefineDNet [56], $D^4$ [53], CUT [32], Santa [51], ODCR [47], and DN [18]. We also include UNSB [22], a Schrödinger Bridge-based image generation method. Notably, Santa, DN, and UNSB were originally designed for unpaired image generation rather than restoration, and we adapt them to dehazing tasks to benchmark their transferability. Following the evaluation protocol of $D^4$, we train all methods on the ITS subset of RESIDE and evaluate on multiple test sets.

**Quantitative Evaluation.** Table 3 summarizes the performance of DDSB under a generalized dehazing protocol. All methods are trained only on the SOTS-indoor dataset and evaluated on three benchmarks: SOTS-indoor, SOTS-outdoor, and NH-HAZE 2. DDSB achieves the best performance across all settings. In terms of PSNR, DDSB outperforms DN by +1.60 dB on SOTS-indoor, +1.49 dB on SOTS-outdoor, and +0.77 dB on NH-HAZE 2. For SSIM, the improvements are +0.009, +0.009, and +0.014, respectively. These results demonstrate DDSB's strong generalization ability to both synthetic and real-world haze conditions. Moreover, DDSB is highly efficient: despite its effectiveness, it only uses 14.68M parameters and runs at 0.019 ms per $512 \times 512$ image—significantly faster than most baselines, including DN (87.7 ms) and RefineDNet (248.5 ms).

**Qualitative Evaluation.** Fig. 4 shows qualitative comparisons. $D^4$ preserves some scene-level consistency but often leaves residual haze and lacks fine detail recovery. DN introduces texture artifacts and inconsistent brightness in challenging outdoor or real-world scenes. DDSB demonstrates

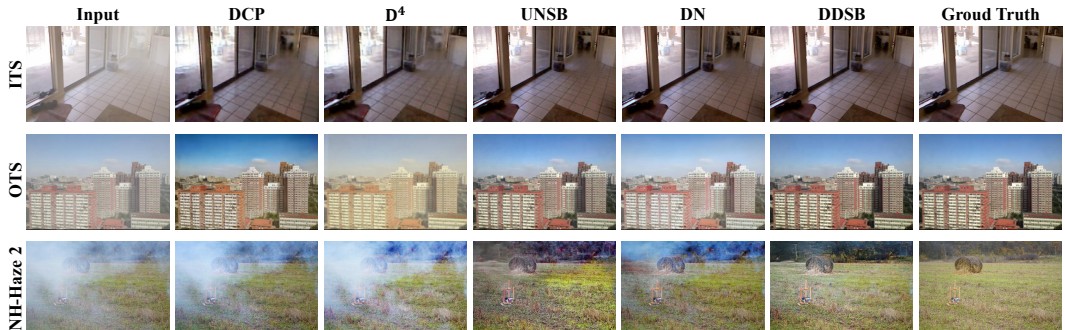

Figure 4: Comparison of state-of-the-art unpaired methods on generalized haze removal.

the most robust generalization among all compared methods. On SOTS-outdoor, it effectively removes haze while maintaining edge sharpness and color balance. On NH-HAZE 2, DDSB delivers perceptually clean outputs with restored textures, faithful color rendering, and minimal artifacts.

## 5.3  Ablation Studies

**Effect of Each Component.** We conduct an ablation study on the SOTS-indoor and SOTS-outdoor datasets. The results are summarized in Table 4. Starting from the baseline equipped with only EROT, we observe consistent improvements when adding either DOT or DTC. Specifically, DOT brings a PSNR gain of +1.55dB on SOTS-indoor and +1.74dB on SOTS-outdoor, as well as SSIM improvements of +0.023 and +0.009, respectively. When adding DTC alone, PSNR improves by +0.43dB on SOTS-indoor and +0.62dB on SOTS-outdoor, and SSIM increases by 0.017 and 0.003, respectively. This shows that DTC improves the intermediate trajectory regularity. Combining DOT and DTC yields the best overall performance, confirming their complementarity.

**Effect of Time Step $N$.** Table 5 analyzes how the number of time steps $N$ in the trajectory affects restoration performance. The default setting is $N = 5$. When $N$ is reduced to 2 or 3, the performance drops notably (e.g., PSNR drops to 26.18dB at $N = 2$). When $N$ increases beyond 5 to 6, the performance slightly degrades, likely due to over-discretization and increased interpolation error.

**Effect of Hyper-parameter $\lambda$.** Table 6 studies the sensitivity of our method to the regularization weight $\lambda$ in Eq. 9, which governs the strength of the entropy-regularized transport. We observe that $\lambda = 0.01$ achieves the highest performance (27.85dB / 0.956). Setting $\lambda$ too low (*e.g.*, 0.0001) degrades performance significantly (down to 25.76dB / 0.940), as the EROT constraint becomes too weak. Conversely, overly large $\lambda$ values (e.g., 1.0) also degrade performance to 26.91dB / 0.946, likely because the transport term dominates and suppresses reconstruction.

**Effect of Layer Depth $D_\phi$.** We conduct an ablation study by varying the number of convolutional layers in $D_\phi$ from 2 to 7. As shown in Table 7, the 3-layer configuration offers the best trade-off between performance and stability. While the 5-layer variant yields a slightly higher SSIM (0.958), its overall PSNR is marginally lower than the 3-layer setting. The 2-layer model underfits the degradation guidance, and the 7-layer version suffers a noticeable performance drop due to training instability.

## 6  Conclusion

In this work, we presented Degradation-aware Dynamic Schrödinger Bridge (DDSB), a novel framework for unpaired image restoration that addresses the limitations of paired data and static restoration dynamics. By leveraging Schrödinger Bridge modeling between unpaired domains and incorporating constraints based on degradation amplification, DDSB enhances the realism of restored outputs. Extensive experiments on diverse degradation scenarios demonstrated the effectiveness and generalizability of our approach, establishing a new direction for principled, data-efficient image restoration in real-world settings.

**Limitation, Future Work & Societal Impact.** DDSB approaches image degradation from a machine learning perspective. Some physical degradation models can be integrated in the future. This work benefits various image degradation scenarios, and we do not envision its negative societal impact.

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

# A    Technical Appendices and Supplementary Material

## A.1    Tighter Generalization Error Bound

In this subsection, we provide a step-by-step deduction of the generalization error bounds for the proposed *Degradation-aware Dynamic Schrödinger Bridge (DDSB)*, and also demonstrate that it has a tighter generalization error bound than the Schrödinger Bridge (SB) baseline.

**Definition 1. Generalization Error.** *The generalization error $\mathcal{E}_{gen}$ quantifies the discrepancy between the predicted clean image $\hat{x}_c$ and the true clean image $x_c$, defined as:*

$$\mathcal{E}_{\text{gen}} = \mathbb{E}\left[\|\hat{x}_c - x_c\|^2\right], \tag{16}$$

*where $\hat{x}_c$ denotes the clean image generated by the model, and $x_c$ denotes the true clean image.*

**Lemma 1. Optimal Transport Formulation.** *The problem of learning the optimal transport plan $\pi^\star$ between the degraded image distribution $\pi_d$ and the clean image distribution $\pi_c$ is formulated as:*

$$\pi^\star = \underset{\pi \in \Pi(\pi_d, \pi_c)}{\arg\min} \ \mathbb{E}_{(x_d, x_c) \sim \pi}\left[\|x_d - x_c\|^2\right] - 2\tau\,\mathcal{H}(\pi), \tag{17}$$

where $\mathcal{H}(\pi)$ denotes the joint entropy of the transport plan $\pi$, and $\tau$ denotes the regularization hyperparameter.

**Proof:** Please refer to [13] for the detailed proof. Briefly, the objective is to minimize the discrepancy between $\pi_d$ and $\pi_c$ while regularizing the transport plan $\pi$ by its entropy. This ensures that the transport process remains smooth and avoids overfitting. The entropy term $\mathcal{H}(\pi)$ penalizes complex transport plans, promoting a smoother and more generalizable solution.

**Theorem 1. Tighter Generalization Error Bound of the Proposed DDSB.** *Compared to the Schrödinger Bridge baseline, the proposed Degradation-aware Dynamic Schrödinger Bridge (DDSB) has a tighter generalization error bound.*

**Proof:** The generalization error is linked to the entropy regularization term $\mathcal{H}(\pi)$. The term $\frac{1}{N}$ reflects the error due to the finite number of transport steps, while the term $\tau$ controls the smoothness of the transport plan. As the number of transport steps increases, the transport plan approximates the true coupling between $\pi_d$ and $\pi_c$, thus reducing the generalization error. Moreover, increasing $\tau$ reduces the impact of entropy regularization, helping to prevent overfitting and further improving generalization. To summarize, the generalization error for the entropy-regularized optimal transport formulation, i.e. Schrödinger Bridge baseline, is bounded as:

$$\mathcal{E}_{\text{gen}}^{\text{SB}} \leq O\left(\frac{1}{N} + \tau\right), \tag{18}$$

where $N$ denotes the number of transport steps, and $\tau$ denotes the temperature hyperparameter.

The DDSB method incorporates the *degradation-aware term $\lambda$*, which helps the model remain consistent with the degradation process during the restoration. This term reduces error accumulation during the iterative restoration process and improves the generalization performance. The total generalization error is composed of three components: 1. The transport error $O\left(\frac{1}{N}\right)$, which decreases as the number of transport steps increases. 2. The entropy regularization error $O(\tau)$, which ensures smooth transport. 3. The degradation fidelity term $O(\lambda)$, which helps the restoration process respect the underlying degradation. Thus, the generalization error for DDSB is bounded as:

$$\mathcal{E}_{\text{gen}}^{\text{DDSB}} \leq O\left(\frac{1}{N} + \tau + \lambda\right). \tag{19}$$

Combining Eq. 18 and Eq. 19, we conclude that:

$$\mathcal{E}_{\text{gen}}^{\text{DDSB}} \leq \mathcal{E}_{\text{gen}}^{\text{SB}} - O(\lambda), \tag{20}$$

demonstrating that the inclusion of the degradation-aware term $\lambda$ results in a tighter error bound for DDSB.

We conclude this subsection by the following remark. This theoretical analysis demonstrates that the proposed DDSB method provides superior generalization performance compared to the SB baseline.

## A.2 Connection to Physical Model based Degradation

In this subsection, we provide a theoretical analysis of the *Degradation-aware Dynamic Schrödinger Bridge (DDSB)*, focusing on its connection to *degradation-aware* techniques and the integration of a *physical degradation model*. This analysis formalizes how the proposed method ensures realistic image restoration by incorporating the degradation process into the transport framework and learning to reverse it.

**Degradation-aware Transport (DOT) and its Role.** The core innovation in DDSB lies in the *Degradation-aware Optimal Transport (DOT)* term. Traditional Schrödinger Bridge (SB) methods focus on learning a transport plan between the degraded and clean image distributions, without considering the physical degradation process that generated the degraded image. DDSB, on the other hand, introduces a *degradation-aware component*, which ensures that the transport process respects the underlying degradation dynamics at every step.

Let $\pi_d$ and $\pi_c$ denote the degraded and clean image distributions, respectively. The objective in DDSB is to learn a transport plan $\pi^\star$ that minimizes the discrepancy between these distributions while respecting the degradation model. The optimal transport cost is modified to include the degradation-aware term:

$$c_{\text{DOT}}(\boldsymbol{x}_d, \boldsymbol{x}_c) = \|\boldsymbol{x}_d - \boldsymbol{x}_c\|^2 + \lambda \cdot \|D_\phi(\boldsymbol{x}_c) - \boldsymbol{x}_d\|^2, \tag{21}$$

where $\boldsymbol{x}_d$ is a sample from the degraded image distribution $\pi_d$, $\boldsymbol{x}_c$ is a clean image sample, $D_\phi(\boldsymbol{x}_c)$ denotes the degraded version of $\boldsymbol{x}_c$ predicted by the learned degradation model $D_\phi$, and $\lambda$ is a hyperparameter controlling the strength of the degradation fidelity term.

The term $\lambda \cdot \|D_\phi(\boldsymbol{x}_c) - \boldsymbol{x}_d\|^2$ enforces that the transport process not only minimizes the discrepancy between $\boldsymbol{x}_d$ and $\boldsymbol{x}_c$, but also ensures that the restoration process is consistent with the learned degradation model $D_\phi$. This degradation-aware term helps maintain physical realism during the restoration process and prevents the generation of unrealistic artifacts.

**Physical Degradation Model $D_\phi$ and its Role.** In real-world image restoration, the degradation process is typically complex, involving factors such as noise, blur, and distortion. In many cases, it is impractical to assume a simple degradation model (such as Gaussian noise). Therefore, DDSB uses a *learnable degradation model $D_\phi$*, which is trained to simulate various degradation processes, such as motion blur, fog, and noise. The model $D_\phi$ learns to predict the degradation process for a given clean image, providing a more accurate representation of real-world degradation dynamics.

The degradation model $D_\phi$ is integrated into the transport process by penalizing discrepancies between the degraded version of the predicted clean image and the degraded input image. Specifically, at each transport step, we introduce a degradation consistency term that ensures the restored image $\boldsymbol{x}_c$ remains aligned with the degradation process at each intermediate step. This term is added to the transport cost as follows:

$$\mathcal{L}_{\text{DOT}} = \mathbb{E}_{q_\theta(\boldsymbol{x}_d)} \left[ \|\boldsymbol{x}_d - \boldsymbol{x}_c(\boldsymbol{x}_d)\|^2 + \lambda \|D_\phi(\boldsymbol{x}_c(\boldsymbol{x}_d)) - \boldsymbol{x}_d\|^2 \right], \tag{22}$$

where $\boldsymbol{x}_d$ is a degraded sample, $\boldsymbol{x}_c(\boldsymbol{x}_d)$ denotes the predicted clean image at a given transport step, and $D_\phi(\boldsymbol{x}_c(\boldsymbol{x}_d))$ denotes the degraded version of $\boldsymbol{x}_c(\boldsymbol{x}_d)$ predicted by the degradation model $D_\phi$.

This ensures that each intermediate image in the restoration trajectory follows the degradation process as closely as possible, improving the physical realism of the restoration.

**Degradation Amplification for Realism.** In this approach, the degradation model $D_\phi$ is not only used to predict the degradation but is also employed to *amplify the degradation* at each step. This means that the model simulates the inverse of the restoration process by artificially degrading the predicted clean image, making the difference between the degraded image $\boldsymbol{x}_d$ and the predicted clean image more pronounced. This helps in preventing the model from generating unrealistic images that would not correspond to any physical degradation.

Formally, the degradation model $D_\phi$ is learned to *amplify* the degradation, i.e., simulate the inverse process of restoration. The objective is to penalize large deviations between the degraded input $\boldsymbol{x}_d$ and the predicted clean image $\boldsymbol{x}_c$ by introducing a degradation consistency term, as described in the DOT loss:

$$\|D_\phi(\boldsymbol{x}_c) - \boldsymbol{x}_d\|^2, \tag{23}$$

which ensures that the model's restoration trajectory remains consistent with the degradation process, leading to more realistic results.

**Integrating the Degradation Model into the Transport Process.** The key idea behind DDSB is to integrate the *degradation model $D_\phi$* into the *optimal transport* framework. The method learns a transport plan $\pi^\star$ that minimizes the distance between the degraded and clean images while ensuring that intermediate steps in the restoration process respect the physical degradation dynamics. This is achieved by adding the degradation-aware term to the transport cost.

At each step of the restoration, we aim to *align the transport plan with the degradation process*. The total loss for DDSB can be expressed as the sum of the standard SB loss and the degradation-aware term:

$$\mathcal{L}_{\text{DDSB}} = \mathcal{L}_{\text{SB}} + \mathcal{L}_{\text{DOT}}, \tag{24}$$

where $\mathcal{L}_{\text{SB}}$ denotes the standard Schrödinger Bridge loss (transport loss). In addition, $\mathcal{L}_{\text{DOT}}$ denotes the degradation-aware optimal transport loss, which ensures that the restoration process respects the degradation model $D_\phi$.

This dual loss function encourages the model to minimize the transport cost while ensuring that the restored images are consistent with the degradation process, leading to more realistic and physically plausible restoration results.

We conclude this subsection by the following remark. The introduction of the degradation-aware term $\lambda$ in the DOT loss improves the generalization of the restoration model. By enforcing consistency with the degradation process, DDSB ensures that the model can generalize well to unseen degradation types. This also reduces the risk of generating unrealistic artifacts, which is a common problem in unpaired image restoration tasks.

The *Degradation-aware Dynamic Schrödinger Bridge (DDSB)* method introduces a novel integration of *degradation-aware optimal transport* and a *physical degradation model*. By incorporating the degradation model $D_\phi$ into the transport process, DDSB ensures that the restoration trajectory remains consistent with the physical degradation process, improving the realism of the restored images. This is achieved by penalizing discrepancies between the degraded input and the predicted clean images, leading to a more physically plausible restoration process that reduces artifacts and improves generalization to unseen degradation types.

This theoretical analysis, along with the introduced degradation-aware and dynamic transport components, lays the foundation for DDSB's superior performance in unpaired image restoration, addressing the key challenges of realism and generalization.

# B  More Visual Results

## B.1  On Multi-Task Image Restoration

More visual results of multi-task image restoration are provided in Fig. 5 and Fig. 6.

## B.2  On Generalized Haze Removal

More visual results of generalized haze removal are provided in Fig. 7 and Fig. 8.

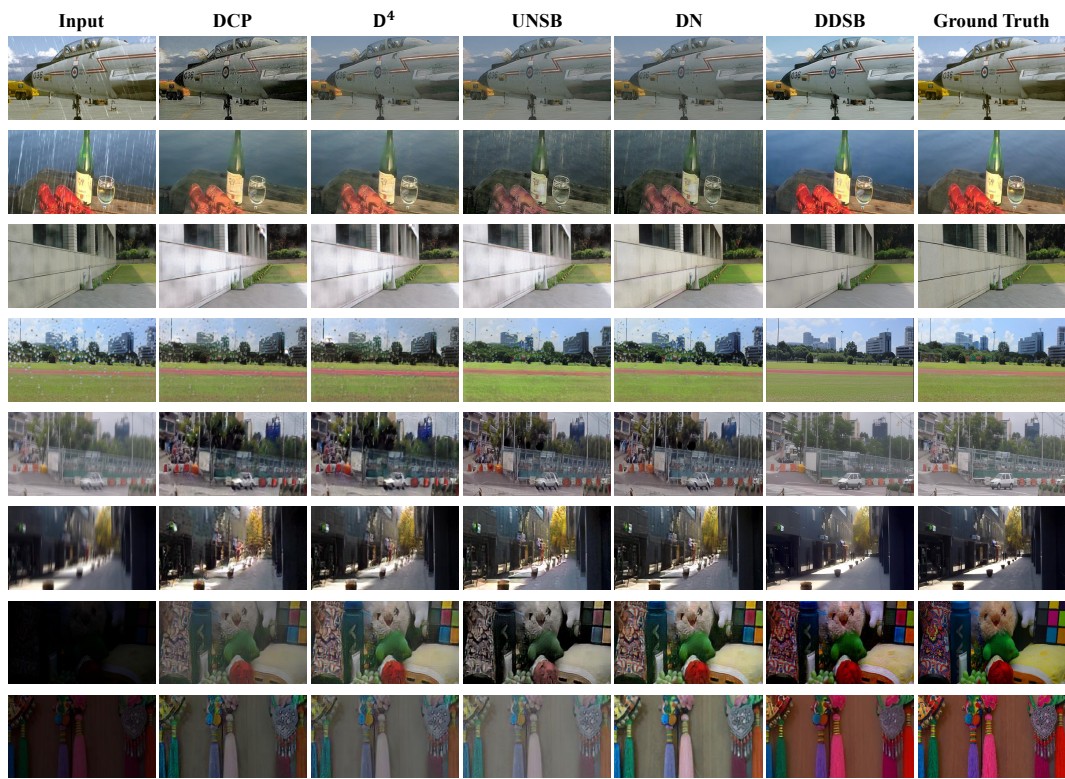

Figure 5: More results of multi-task image restoration

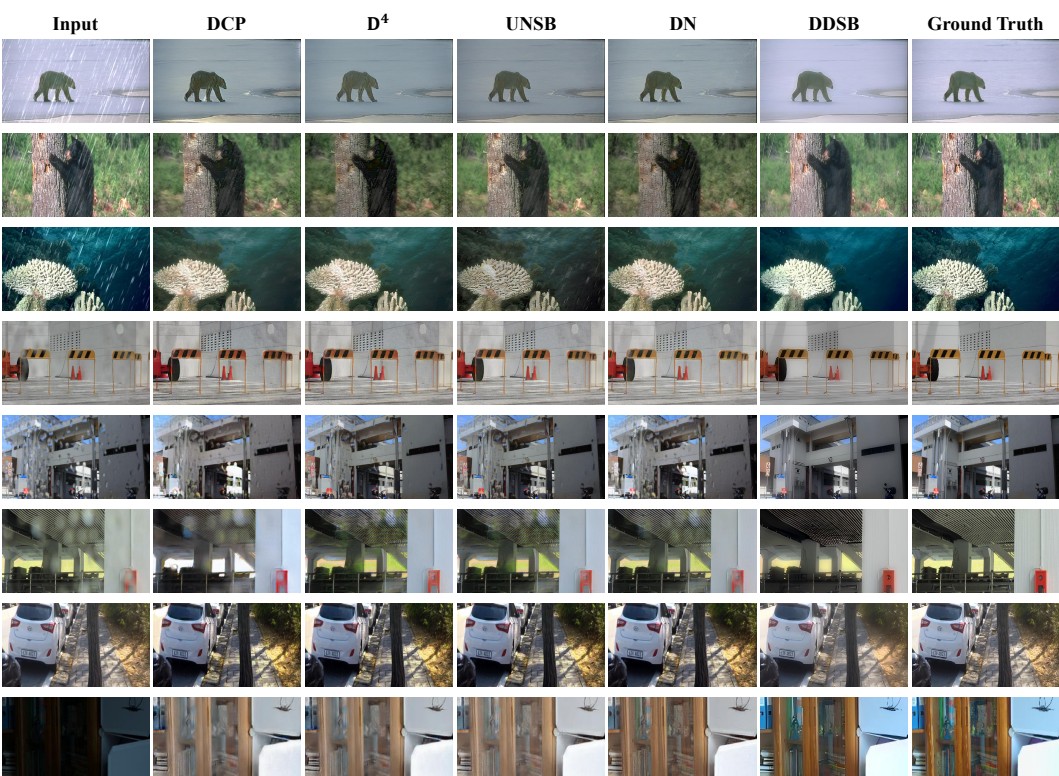

Figure 6: More results of multi-task image restoration

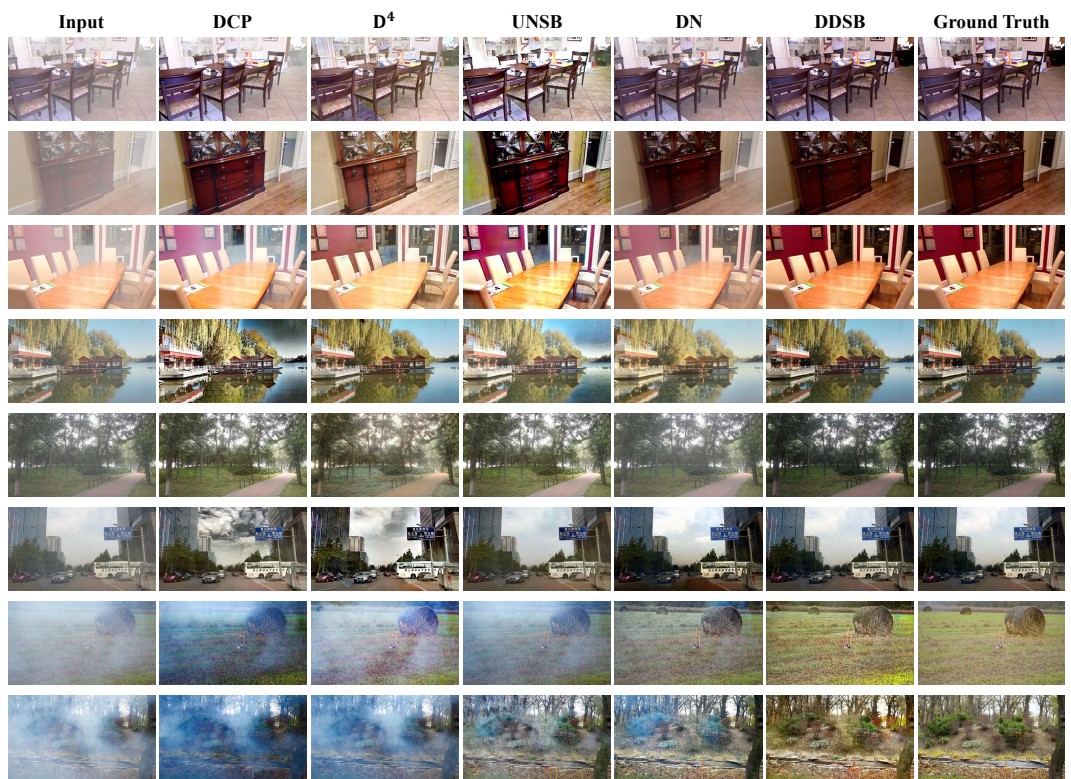

Figure 7: More results of generalized haze removal

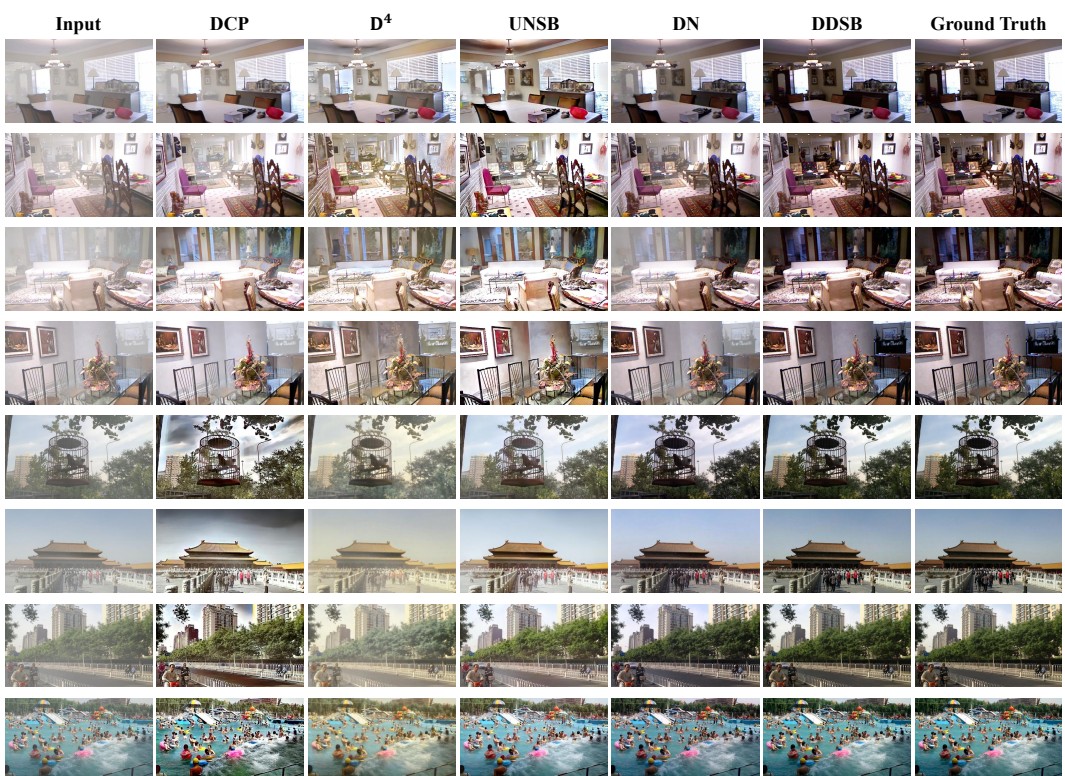

Figure 8: More results of generalized haze removal

