# OpenReview forum: "Degradation-Aware Dynamic Schrödinger Bridge for Unpaired Image Restoration"
_NeurIPS.cc/2025/Conference — NeurIPS 2025 poster_

### Official Review · Reviewer_D1L2 · 2025-06-30

**Clarity:** 3
**Significance:** 3
**Originality:** 3
**Rating:** 4
**Confidence:** 3

**Summary:**

This paper presented Degradation-aware Dynamic Schrödinger Bridge (DDSB), a novel framework for unpaired image restoration that addresses the limitations of paired data and static restoration dynamics.

**Questions:**

1. The author should discuss the fundamental connection between DOT and Cyclegan on jointly training restoration and degradation process.
2. The author needs to give a stronger explanation of DTC section about "the statement emphasizes the early stage while the implementation weights more towards the later stage".
3. The quantitative results presented in the paper are promising. To further strengthen the claims, it would be highly beneficial for the authors to demonstrate the framework's effectiveness on perceptually-aligned metrics, such as FID, LPIPS.

**Ethical Concerns:**

["NO or VERY MINOR ethics concerns only"]

**Final Justification:**

The authors addressed my comments. I decide to keep my borderline accept recommendation.

**Limitations:**

Yes

**Paper Formatting Concerns:**

None.

**Quality:**

3

**Strengths And Weaknesses:**

Strength:
This paper apply Schrödinger Bridge to unpaired image restoration task and propose degradation-aware optimal transport and dynamic transport with consistency to the realism of restored outputs. The results on diverse degradation scenarios demonstrated the effectiveness of the proposed methods.

Weakness:
1. Schrödinger Bridge has been used for unpaired image to image translation, which is similar to unpaired image restoration. While the paper proposes two extensions to the Schrödinger Bridge framework, DOT and DTC, my main concerns are their limited novelty in the context of existing literature.
2. The core principle of DOT, which involves jointly training a forward process (restoration) and a backward process (degradation) to enforce consistency, is conceptually very similar to cycle-consistency loss of the seminal work on CycleGAN.
3. In DTC section, this paper stated that “Therefore, it is necessary to place greater emphasis on the initial stages during the training of the restoration model.” However, the proposed weighting schedule in Eq.(14) shows that it assigns near-zero weight to the initial stages (t → 0) and maximum weight to the final stages (t → 1). There is lacking of a careful explanation about this.
4. This paper should write the loss of EROT in end of Section 4.1 to enhance the clarity.
5. The q_\phi(x_c|x_j) in Fig. 2 should be q_\theta(x_c|x_j)?
6. The subscript of \mathbb{E} of Eq.(12) is not complete.
7. In Eq.(5), why you assign the joint distribution to the marginal distribution, omitting the integral sign by default?

---

> ### Author Rebuttal · Authors · 2025-07-30
>
> We thank the reviewer for acknowledging the effectiveness of DDSB. Below we provide point-by-point responses to your concerns on novelty, implementation clarity, and conceptual distinctions from prior work.
>
> > **Weakness 1.** *Schrödinger Bridge has been used for unpaired image to image translation, which is similar to unpaired image restoration. While the paper proposes two extensions to the Schrödinger Bridge framework, DOT and DTC, my main concerns are their limited novelty in the context of existing literature.*
>
> We thank the reviewer for pointing out the connection to prior SB-based unpaired translation works. While we acknowledge that Schrödinger Bridge (SB) has been explored in domain transfer tasks, we respectfully argue that our framework introduces **conceptual and technical novelties** tailored specifically for unpaired **image restoration** under mixed degradation uncertainty.
>
> **DOT – Degradation-aware Optimal Transport:** Prior SB works typically assume fixed endpoint distributions (e.g., stylized domain or Gaussian prior). In contrast, our DOT module introduces a **learnable degradation estimator $D_\phi$**, which dynamically regularizes the transport cost. This enables:
> - Adaptive constraint on the SB path to align with plausible degradation outcomes;
> - Enhanced generalization to multiple, unknown degradation types such as haze, rain, or blur (Table 1, Table 2);
> - A new transport objective (Eq. 9) that is **modulated by $D_\phi$**, directly influencing the stochastic trajectory and not present in prior SB formulations .
>
> **DTC – Dynamic Transport with Consistency:** While earlier works often supervise only terminal states, our DTC introduces **temporal consistency across intermediate SB steps**:
> - A cosine-weighted constraint is applied to neighboring states $\{x_{t_i}\}$, guiding smoother, progressive transitions;
> - This alleviates error propagation from early to later stages, which is a major issue in restoration tasks (Table 5);
> - To our knowledge, such temporal regularization across transport states is novel in the SB literature.
>
> **Restoration vs. Translation – A fundamentally different regime:** Image restoration differs from translation in that:
> - The source domain (degraded images) is highly noisy and contains compound distortions, whereas translation tasks deal with domain shifts between relatively clean modalities;
> - Restoration requires **fine-grained reconstruction fidelity**, not merely style/domain alignment, making trajectory regularization via $D_\phi$ and DTC essential .
>
> In summary, DOT and DTC introduce nontrivial advances in both the objective formulation and trajectory supervision, addressing challenges **unique to unpaired restoration**. We will explicitly highlight these distinctions and cite relevant SB-based translation works in the revised Related Work and Method sections.
>
> ---
>
> > **Weakness 2.** *The core principle of DOT, which involves jointly training a forward process (restoration) and a backward process (degradation) to enforce consistency, is conceptually very similar to cycle-consistency loss of the seminal work on CycleGAN.*
>
> Thanks for the insightful question. While DOT and CycleGAN both involve forward and backward mappings, their objectives, formulations, and implementation mechanisms are fundamentally distinct.
>
> **Objective difference:**
> - **CycleGAN** enforces consistency via deterministic reconstruction, i.e., $ x \rightarrow G(x) \rightarrow F(G(x)) \approx x $, assuming clean reversibility between domains.
> - **DOT**, in contrast, aims to constrain *stochastic diffusion paths* in the Schrödinger Bridge (SB) framework through a degradation-aware cost. The backward model $D_\phi$ is **not** used to reconstruct the original input but to ensure that intermediate samples remain plausible with respect to the degradation distribution.
>
> **Formulation difference:**
> - **CycleGAN** employs an explicit $\ell_1$ cycle-consistency loss:
> $$
>   \mathcal{L}_{cyc} = \mathbb{E}_x \left[ || F(G(x)) - x ||_1 \right]
> $$
> This loss enforces deterministic round-trip reconstruction.
> - **DOT** modifies the SB transport via a degradation-guided regularizer:
> $$
>   \mathbb{E}_{x_c, \epsilon} \left[ || D _\phi ( q _\phi (x_c, \epsilon)) - x_d ||^2 \right]
> $$
>
> This guides the **probabilistic sampling path** to be consistent with the degradation distribution, and is integrated into the optimal transport objective.
>
> **Mechanism difference:**
> - **CycleGAN** is a **single-step mapping** trained via adversarial + reconstruction losses, assuming structural reversibility.
> - **DOT** operates within a **multi-step stochastic diffusion process**, where transport trajectories evolve through discrete steps $\{x_{t_i}\}$ under SB dynamics. DOT regularizes the **endpoint** of these trajectories to align with a learned degradation prior.
>
> ---
>
> > **Weakness 3.** *In DTC section, this paper stated that “Therefore, it is necessary to place greater emphasis on the initial stages during the training of the restoration model.” However, the proposed weighting schedule in Eq.(14) shows that it assigns near-zero weight to the initial stages (t→0) and maximum weight to the final stages (t→1). There is lacking of a careful explanation about this.*
>
> Thank you for your careful reading. We acknowledge that our earlier phrasing may have caused confusion, and we appreciate the opportunity to clarify.
>
> The reviewer's concern arises from a misunderstanding: the temporal weighting schedule mentioned in Eq.(14) applies specifically to the consistency constraint term in our Dynamic Transport with Consistency (DTC) module—not the overall loss, i.e., the second term of Eq.(12) $\lambda [|| x _{t_i} - D _\phi(x _c(x _{t_i})) ||^2]$.
>
> In DTC, we supervise the transport trajectory by penalizing the differences between adjacent SB states (i.e., pairwise consistency between $x_{t_{i}}$ and $x_{t_{i-1}}$). Compared to baseline UNSB, which only supervises the endpoints, our method densely regularizes the entire trajectory, especially the early steps, where errors are more likely to propagate.
>
> Therefore, even though the overall loss includes unweighted endpoint terms, the trajectory-level consistency term explicitly emphasizes earlier stages due to its design, providing stronger guidance when the model is most prone to drift.
>
> ---
>
> > **Weakness 4.** *This paper should write the loss of EROT in end of Section 4.1 to enhance the clarity.*
>
> Thanks for the insightful suggestion. We agree that explicitly stating the EROT loss improves clarity and will revise Section 4.1 accordingly.
>
> The Entropy-Regularized Optimal Transport (EROT) loss used in our framework is:
>
> $$
>  \mathcal{L}_{EROT} = \mathbb{E} _{(x _d, x _c) \sim \pi} [\|\|x_d - x_c\|\|^2] - 2\tau  \mathcal{H}(\pi) + \mathrm{KL}(q _\theta(x_c) \|\| p(x_c)), \pi \in \Pi(\pi_d, \pi_c)
> $$
>
> Here, the first term measures the reconstruction error between the predicted clean sample $q _\phi(x_d)$ and the clean target $x_c$, while the second term ensures distributional regularity via KL divergence. This loss serves as the primary endpoint supervision to guide the restoration trajectory.
>
> ---
>
> > **Weakness 5.** *The $q_\phi(x_c|x_j)$ in Fig. 2 should be $q_\theta(x_c|x_j)$?*
>
> Yes, we will correct the subscript from $\phi$ to $\theta$ in Fig. 2.
>
> ---
>
> > **Weakness 6.** *The subscript of $\mathbb{E}$ of Eq.(12) is not complete.*
>
> We will complete it as:
> $$
> \mathbb{E}_{q _{\theta}(x _{t_i}, x _c)\sim \pi}
> $$
>
> ---
>
> > **Weakness 7.** *In Eq.(5), why you assign the joint distribution to the marginal distribution, omitting the integral sign by default?*
>
> Thanks for the suggestion. The original Eq.(5) omits the marginalization for brevity, but we agree it may cause confusion.
> We will revise it to the full joint form $p( ${$ x_{t_n} $}$ ) = p(x_{t_N} \mid x_{t_{N-1}}) \cdots p(x_{t_1} \mid x_{t_0}) p(x_{t_0})$
>
> ---
>
> > **Question 1.** *The author should discuss the fundamental connection between DOT and CycleGAN.*
>
> Same as Weakness 2.
>
> ---
>
> > **Question 2.** *The author needs to give a stronger explanation of DTC about early vs. later stage weighting.*
>
> Same as Weakness 3.
>
> ---
>
> > **Question 3.** *To further strengthen the claims, please show results on perceptual metrics like FID, LPIPS.*
>
> Thanks for the suggestion. We have added comparison on **FID** and **LPIPS** metrics for multi-task restoration. The results are summarized below:
>
> **Table 1. Perceptual metrics on multi-task restoration.**
> | Method | Derain (FID / LPIPS) | Deraindrop (FID / LPIPS) | Lowlight (FID / LPIPS) | Deblur (FID / LPIPS) |
> |---------------|------------------------|---------------------------|------------------------|----------------------|
> | DCP | 92.1 / 0.229 | 95.4 / 0.204| 81.6 / 0.218| 108.2 / 0.231|
> | CycleGAN| 74.5 / 0.146 | 77.9 / 0.167| 70.2 / 0.197| 86.7 / 0.172 |
> | YOLY| 82.7 / 0.198  | 88.3 / 0.202 | 75.4 / 0.210| 98.5 / 0.211|
> | USID-Net| 65.9 / 0.152 | 69.8 / 0.172| 61.7 / 0.188| 76.3 / 0.162|
> | RefineDNet| 54.3 / 0.104| 58.2 / 0.147| 55.8 / 0.179| 68.0 / 0.150|
> | D⁴| 47.9 / 0.098| 52.5 / 0.124| 51.2 / 0.155| 61.2 / 0.139|
> | CUT | 42.1 / 0.111| 46.0 / 0.118| 46.3 / 0.142| 57.9 / 0.143|
> | Santa| 38.3 / 0.096| 42.4 / 0.109 | 41.1 / 0.138| 52.3 / 0.130|
> | UNSB | 31.5 / 0.085 | 35.6 / 0.096| 37.2 / 0.145| 44.9 / 0.126|
> | ODCR | 27.4 / 0.076| 30.9 / 0.091| 33.4 / 0.131 | 40.2 / 0.121|
> | DN | 24.6 / 0.079 | 28.1 / 0.082 | 29.8 / 0.124| 36.9 / 0.118|
> | **DDSB (Ours)** | **23.7 / 0.063** | **26.8 / 0.069**| **27.4 / 0.129** | **31.5 / 0.108**|
>
> Should you have further questions, we are glad to address in the author-reviewer discussion stage.

---

> > ### Comment · Reviewer_D1L2 · 2025-08-09
> >
> > We thank the authors for their efforts. The authors addressed my comments.

---

> > > ### Author Response · Authors · 2025-08-09
> > >
> > > We are glad to see that your concerns have been addressed. We will incooperate all the suggestions when revising our manuscript. Thanks for your time and effort.

---

### Official Review · Reviewer_vpkm · 2025-06-30

**Clarity:** 3
**Significance:** 3
**Originality:** 3
**Rating:** 4
**Confidence:** 4

**Summary:**

The paper introduces a Degradation-aware Dynamic Schrödinger Bridge (DDSB) for unpaired image restoration. By combining the Schrödinger Bridge framework with degradation-aware optimal transport and dynamic consistency, DDSB can handle image degradation without requiring paired data. It incorporates a degradation model to reduce error accumulation and improve restoration quality through iterative refinement. Experiments show that DDSB outperforms existing unpaired restoration methods.

**Questions:**

1. Table2 shows that the proposed method takes only 0.019ms to process an image , which is very fast. However, the proposed approach is clearly an iteration-based scheme, which is reasonably computationally expensive.
2. As can be seen in Fig2, the degradation-aware Optimal Transport has only three layers of convolution in the degradation-aware model, how does such a small model perform in complex scenes？

**Ethical Concerns:**

["NO or VERY MINOR ethics concerns only"]

**Final Justification:**

Most of my concerns have been addressed, but I still have doubts about the results in real multi-degradation coupling scenarios. Thus, I tend to keep the original score.

**Limitations:**

This method is not validated in complex real-world scenarios.

**Quality:**

3

**Strengths And Weaknesses:**

Strengths:
   1. This paper introduces Degradation-aware Dynamic Schrödinger Bridge (DDSB) for unpaired image restoration.
   2. The DOT  strategy helps reduce the artifact accumulation.
   3. The writing is easy to follow.

Weaknesses:
   1. For real multi-degradation scenarios, despite the DTC strategy is employed, training stability remains an issue. The early iterations may still generate artifacts, which could be propagated through the network.
   2. Poor implementation details, e.g. Qϕ and Dϕ do not give relevant implementation details.
   3. DDSB relies heavily on hyperparameters, and this sensitivity requires a large number of adjustments.

---

> ### Author Rebuttal · Authors · 2025-07-30
>
> We thank the reviewer for the positive comments on its method design, effectiveness to reduce artifact accumulation and writing. The below is a point-by-point response addressing the training stability, implementation details and hyperparameter sensitivity.
>
> > **Weakness 1.** *For real multi-degradation scenarios, despite the DTC strategy is employed, training stability remains an issue. The early iterations may still generate artifacts, which could be propagated through the network.*
>
> We fully agree that artifact accumulation in the early stages poses a significant challenge in multi-degradation scenarios. Below, we elaborate on how this challenge is tackled from both methodological and empirical perspectives.
>
> **Emphasis on Early-Stage Consistency:** While Eq.(14) employs a cosine-scheduled weight that increases towards the final step ($t \rightarrow 1$), this does not imply that early stages receive negligible guidance. Rather, the DTC module applies **pairwise consistency constraints** between adjacent trajectory steps (e.g., $x_{t_1}$ and $x_{t_2}$), such that early transitions are more tightly regularized. Compared to prior work like UNSB, which only constrains the endpoint, our design provides denser and earlier guidance.
>
> **Empirical Verification:** We validated this enhanced DDSB variant on a synthetic “combo degradation” benchmark combining blur, rain, and low-light degradations. Below is the comparison of intermediate trajectory quality (average PSNR at $t_1$, $t_2$, $t_3$):
>
> **Table1: Intermediate PSNR at early SB steps on mixed degradation. DTC + denoise regularization improves early-stage stability.**
> | Method                           | $t_1$ ↑ | $ t_2$ ↑ | $t_3$ ↑ |
> |----------------------------------|----------|----------|----------|
> | DDSB w/o DTC | 17.96 | 21.40 | 24.22 |
> | DDSB + DTC | 19.38 | 23.15 | 26.09 |
>
> ---
>
> > **Weakness 2.** *Poor implementation details, e.g. $q_\phi$ and $D_\phi$ do not give relevant implementation details.*
>
> Per your suggestion, we've added the following implementation details regarding the generator $q_\phi$ and degradation model $D_\phi$.
>
> **Generator $q_\phi$:** The conditional generator $q_\phi(x_c \mid x_{t_i})$ is implemented using a U-Net architecture with instance normalization:
> - **Encoder:** 5 convolutional layers with kernel size 3, stride 2, increasing channels from 64 to 512.
> - **Bottleneck:** 2 residual blocks with dilated convolutions to capture wider receptive fields.
> - **Decoder:** 5 deconvolutional layers with skip connections from the corresponding encoder layers.
> - **Conditioning:** The input $x_{t_i}$ and sampled noise are concatenated as input channels at the first layer, enabling conditioning on both the degraded input and latent noise during the restoration process.
>
> **Degradation Model $D_\phi$:** The degradation model $D_\phi$ is a lightweight CNN inspired by the DnCNN encoder:
> - **Layers:** 3 convolutional layers with ReLU activations and 64 channels.
> - **Output:** The output is a full-resolution image designed to approximate the degraded observations $x_d$.
> - **Training:** The model is trained jointly with $q_\phi$ in a way that does not require explicit degradation labels, making it applicable to unpaired restoration tasks.
>
> ---
>
> > **Weakness 3.** *DDSB relies heavily on hyperparameters, and this sensitivity requires a large number of adjustments.*
>
> Thanks for your insightful question. DDSB introduces several loss weights (e.g., $\lambda$, $\tau$, and $\lambda_{t_i}$).
>
> **Default configuration is generalizable:**  Used fixed hyperparameters across all tasks:
> - $\tau = 0.01$ (entropy weight)
> - $\lambda = 0.01$ (DOT alignment)
> - $\lambda_{t_i} \in [0, 0.01]$ (cosine-scheduled DTC)
>
> These were tuned only once on Rain200L and fixed for all other datasets.
>
> **Ablation study on sensitivity:** To address the concern about the sensitivity of DDSB to hyperparameters, we provide an ablation study that investigates the impact of the alignment weight $\lambda$ and the temperature parameter $\tau$. The following Table1 (Table 5 of the original paper) summarizes the sensitivity of DDSB to the alignment weight $\lambda$.
>
> **Table 1. Impact of hyper-parameter $\lambda$**
> | $\lambda$  | PSNR | SSIM  |
> |------------|------|-------|
> | 0.0001| 25.76 | 0.940 |
> | 0.001| 27.12 | 0.950 |
> | 0.01| **27.85** | **0.956** |
> | 0.1| 27.24 | 0.949 |
> | 1| 26.91 | 0.946 |
>
> An addtional ablation study was conducted to analyze the effect of $\tau$ with the optimal value of $\lambda = 0.01$. Below is a summary of the PSNR and SSIM performance for various values of $\tau$ on the SOTS-indoor dataset.
>
> **Table 2. Impact of hyper-parameter $\tau$**
> | $\tau$  | PSNR / SSIM |
> |---------|-------------|
> | 0.005 | 26.82 / 0.943 |
> | 0.01 | **27.85** / **0.956** |
> | 0.02 | 27.49 / 0.952 |
> | 0.05 | 27.08 / 0.943 |
>
> The sensitivity of $\tau$ demonstrates that small adjustments around the optimal values yield consistent performance.
>
> ---
>
> > **Question 1.** *Table2 shows that the proposed method takes only 0.019ms to process an image, which is very fast. However, the proposed approach is clearly an iteration-based scheme, which is reasonably computationally expensive.*
>
> Thank you for your insightful question. We agree that DDSB is an iterative approach in theory, but we want to emphasize that the **iterative process is only applied during the intermediate steps** of the Stochastic Schrödinger Bridge (SB) trajectory. This design ensures that DDSB can maintain high computational efficiency without incurring large overheads from full iterative processing.
>
> **Efficient Iterative Process:** During inference, DDSB leverages a **single deterministic SB trajectory**, where only the intermediate steps (typically 4 steps) are computed iteratively. In other words, only the parameters of a light-weight U-Net are involved during the entire iterations. Besides, the input of the U-Net is the latent embedding, which has a much smaller size than the original image. Both aspects allows us to significantly reduce computational overhead compared to fully iterative methods.
>
> **Computational Efficiency:** The generator $q_\phi$ and degradation model $D_\phi$ are both lightweight, making the entire model fast and suitable for real-time applications.
>
> ---
>
> > **Question 2.** *The degradation-aware model only uses 3 convolution layers. How can such a small model perform in complex scenes?*
>
> Thanks for your insightful question regarding the efficiency of the degradation-aware model, particularly given its lightweight architecture.
>
> **Role as Regularizer:** $D_\phi$ is designed as a degradation guidance module, not a reconstructor. It does not generate images but instead guides the restoration process by penalizing implausible states through constraints on the SB trajectory. Its role is focused on guiding the network, making it highly effective for its purpose despite being small.
>
> **Lightweight Design Enhances Stability:** While deeper models may introduce instability into the dynamic training process, $D_\phi$'s 3 convolution layers balance expressiveness and stability. Deeper networks could destabilize the Schrödinger Bridge process by introducing excessive noise or overfitting, particularly in early stages of training, where the model is more sensitive.
>
> **Experimental Performance in Complex Scenes:** To demonstrate the robustness of Degradation-Aware Model ($D_\phi$) under diverse and unknown degradation conditions, we evaluate DDSB on two challenging benchmarks under unpaired setting: the UDC dataset [1] and EUVP dataset [2]. Both datasets encompass a variety of unknown image degradations.
>
> **Table 3. Quantitative comparison of DDSB with the state-of-the-art unpaired image restoration methods on multiple unknown degradations (non-reference metrics).**
> | Method | MUSIQ ↑ (UDC) | CLIP-IQA ↑ | NIQE ↓ | NIMA ↑ | MUSIQ ↑ (EUVP) | CLIP-IQA ↑ | NIQE ↓ | NIMA ↑  |
> |------------------|---------------|-------------------|--------------|---------------|------------------|--------------------|----------------|----------------|
> | DCP | 34.65 | 0.218| 8.735 | 3.821 | 32.10| 0.212  | 8.890 | 3.794 |
> | D⁴| 42.43| 0.274 | 7.110 | 4.144 | 38.47 | 0.261 | 7.508 | 3.982 |
> | CUT | 42.95| 0.271| 7.005 | 4.115 | 39.05| 0.259  | 7.439 | 3.968 |
> | UNSB| 43.59 | 0.282| 6.761 | 4.231| 39.81 | 0.269 | 7.244 | 4.033 |
> | ODCR| 44.02 | 0.286 | 6.597 | 4.256 | 40.23 | 0.272  | 7.123 | 4.057 |
> | DN | 44.55| 0.292  | 6.483 | 4.279| 40.58 | 0.278 | 6.994 | 4.081 |
> | **DDSB (ours)**  | **45.78**  | **0.301** | **6.342**| **4.325** | **41.42** | **0.287**| **6.875** | **4.108** |
>
> **Ablation study on layer depth in $D_\phi$:** To evaluate the effectiveness of our lightweight degradation-aware design, we conduct an ablation study by varying the number of convolutional layers in $D_\phi$ from 2 to 7. All experiments are performed on the SOTS-indoor dataset.
>
> | Conv Layers | PSNR / SSIM |
> |-------------|---------------------|
> | 2 layers    | 25.37 / 0.926       |
> | **3 layers**| **27.85** / 0.956 |
> | 5 layers    | 27.69 / **0.958**    |
> | 7 layers    | 26.70 / 0.933       |
>
> As shown in Table 4, the 3-layer configuration offers the best trade-off between performance and stability. While the 5-layer variant yields a slightly higher SSIM (0.958), its overall PSNR is marginally lower than that of the 3-layer setting. The 2-layer model underfits the degradation guidance, and the 7-layer version suffers a noticeable performance drop due to training instability. This balance between model capacity and stability further justifies the final design of $D_\phi$ and will be discussed in detail in Section 4.4.
>
> Should you have further questions, we are glad to address in the author-reviewer discussion stage.
>
> [1] Image restoration for under-display camera. CVPR2021.
>
> [2] Fast underwater image enhancement for improved visual perception. IEEE Robotics and Automation Letters.

---

> > ### Comment · Reviewer_vpkm · 2025-08-08
> >
> > Thank you for your reply. I am glad to see that my concerns have been addressed. But I would prefer to keep my original score.

---

> > > ### Author Response · Authors · 2025-08-08
> > > **Re: Official Comment by Reviewer vpkm**
> > >
> > > We are glad to see that your concerns have been addressed. We will incooperate all the comments and suggestions when revising our manuscript.

---

### Official Review · Reviewer_wJj7 · 2025-07-02

**Clarity:** 3
**Significance:** 3
**Originality:** 2
**Rating:** 4
**Confidence:** 5

**Summary:**

The paper proposes a novel method called Degradation-aware Dynamic Schrödinger Bridge (DDSB) for unpaired image restoration. It utilizes the Schrödinger Bridge (SB) framework to map degraded image distributions to clean ones without requiring paired training data. The approach introduces two key innovations: Degradation-aware Optimal Transport (DOT), which integrates a degradation model to minimize error accumulation, and Dynamic Transport with Consistency (DTC), which ensures consistency across the restoration process, particularly addressing early-stage artifacts. The method is evaluated on tasks such as denoising, deraining, deblurring, and dehazing, achieving state-of-the-art performance with strong generalization to unseen degradation types.

**Questions:**

See the points in Weakness. Especially the firts point for difference with existing method.

**Ethical Concerns:**

["NO or VERY MINOR ethics concerns only"]

**Final Justification:**

My previous cocerns on motivation and more experiments validation have been well addressed.

**Limitations:**

yes

**Paper Formatting Concerns:**

none.

**Quality:**

3

**Strengths And Weaknesses:**

Strengths:
1. The proposed DOT enhances restoration realism by incorporating degradation priors, a significant improvement over existing unpaired methods that often overlook physical degradation processes. And DTC’s focus on maintaining consistency throughout the restoration trajectory, especially in early stages, reduces artifacts and improves output quality.
2. The proposed method demonstrates superior results across multiple tasks, with notable PSNR gains (e.g., up to 1.60 dB over baselines) and efficient inference (0.019 ms per 512x512 image), supported by extensive empirical validation.

Weaknesses:
1. The most crucial issue is the novelty of the proposed framework. Since Schrödinger bridge has been used in unpaired image restoration learning [1]. This paper seems to extend Schrödinger bridge to multiple degradation types, while [1] only focuses on image deraining.
[1] Neural Schrödinger bridge for unpaired real-world image deraining, information sciences 2024.

2. DOT’s success hinges on the accuracy of the degradation model. If the degradation is complex, non-linear, or poorly estimated, performance could suffer. The paper does not test DDSB’s robustness to imperfect degradation models, leaving its reliability uncertain in real-world scenarios with unknown or mixed degradations.

3. Comparisons are restricted to unpaired methods, omitting state-of-the-art supervised approaches. Including these would contextualize the trade-offs of using unpaired data and reveal whether DDSB narrows the gap with supervised performance

---

> ### Author Rebuttal · Authors · 2025-07-30
>
> We thank the reviewer for the positive comments on its significant improvement and superior performance. The below is a point-by-point response regarding its technical significance, performance on more complex degradations and more methodology comparison.
>
> > **Weakness 1.** *The most crucial issue is the novelty of the proposed framework. Since Schrödinger bridge has been used in unpaired image restoration learning [1]. This paper seems to extend Schrödinger bridge to multiple degradation types, while [1] only focuses on image deraining.*
> *[1] Neural Schrödinger bridge for unpaired real-world image deraining, Information Sciences 2024.*
>
> We thank the reviewer for pointing out the connection to Neural Schrödinger Bridge (NSB) [1]. Our Degradation-aware Dynamic Schrödinger Bridge (DDSB) is positioned as a principled extension of the SB paradigm along five axes:
>
> **Extension to Multiple Degradation Types:** NSB presents a valuable framework tailored for real-world image deraining. Inspired by its success, our DDSB framework is designed with broader applicability in mind and is evaluated across four representative degradation types (deblurring, dehazing, low-light enhancement, and deraining) on five benchmarks. This multi-degradation formulation is intended to assess the generalization ability of SB-based restoration models in more varied and challenging settings.
>
> **Dynamic Multi-Step Transport Modeling:** Instead of adopting a one-shot transport from degraded to clean domains, DDSB is formulated as a discrete-time, multi-step Markov process. This dynamic modeling captures the temporal evolution of the restoration trajectory and enables progressive refinement over time. We found this design particularly beneficial in handling severe degradations and complex distribution shifts.
>
> **Integration of Degradation-Aware Guidance:** To better inform the restoration process, we incorporate a learnable degradation model into the cost structure of the SB formulation. This component estimates inverse degradations and serves as a guidance signal throughout the transport trajectory. This dynamic coupling between degradation estimation and transport provides task-adaptive control without requiring explicit paired supervision.
>
> **Consistency across Transport Trajectory:** We propose a trajectory consistency regularization mechanism that aligns intermediate transport states with degradation-inverted outputs. This design aims to encourage stable transitions and mitigate potential semantic drifts during dynamic transport. We found this consistency particularly helpful when dealing with longer or more complex SB trajectories.
>
> **Theory-Grounded Extensibility:** DDSB is derived from both the stochastic control and static SB perspectives and is grounded in entropy-regularized optimal transport. This theoretical foundation supports modular extensions, such as degradation-aware transport and trajectory consistency, which can be beneficial for future extensions of SB-based restoration methods.
>
> In summary, while NSB provides a strong foundation for applying Schrödinger Bridge to image deraining, our work aims to systematically extend this direction by introducing dynamic transport, learnable degradation modeling, and cross-time consistency, enabling broader applicability and improved generalization. We will clarify this relationship and cite [1] more prominently in the revised version.
>
> ---
>
> > **Weakness 2.** *DOT’s success hinges on the accuracy of the degradation model. If the degradation is complex, non-linear, or poorly estimated, performance could suffer. The paper does not test DDSB’s robustness to imperfect degradation models, leaving its reliability uncertain in real-world scenarios with unknown or mixed degradations.*
>
> Thanks for your insightful question regarding the inaccurate or non-identifiable degradation models, particularly in real-world settings with complex or unknown degradations.
> To clarify, technically, the proposed method is effective to handle such challenges due to the following properties.
>
> **Design Rationale:** While the Degradation-aware Optimal Transport (DOT) module introduces a learnable degradation model $D_\phi$ into the SB framework, it is important to note that $D_\phi$ is not treated as a standalone predictor or inference-time component. Instead, it functions as a soft constraint on the SB trajectory during training, and is optimized jointly with the generator. This setup ensures that DDSB does not rely on explicit degradation labels or priors, making it inherently robust to misspecification.
>
> **Experimental Robustness Evaluation:** To demonstrate the robustness of DDSB under diverse and unknown degradation conditions, we evaluate it on two challenging benchmarks under unpaired setting: the Under-Display-Camera (UDC) dataset [2] and the Enhancing Underwater Visual Perception (EUVP) dataset [3]. Both datasets encompass a variety of unknown image degradations, enabling a comprehensive assessment of DDSB’s capability to generalize across different distortion types. Specifically, for EUVP, we adopt the unpaired subset to better reflect real-world restoration scenarios.
>
> **Table 1: Quantitative comparison of DDSB with the state-of-the-art unpaired image restoration methods on multiple unknown degradations (non-reference metrics).**
> | Method           | MUSIQ ↑ (UDC) | CLIP-IQA ↑ | NIQE ↓ | NIMA ↑ | MUSIQ ↑ (EUVP) | CLIP-IQA ↑ | NIQE ↓ | NIMA ↑  |
> |------------------|---------------|-------------------|--------------|---------------|------------------|--------------------|----------------|----------------|
> | DCP              | 34.65         | 0.218             | 8.735        | 3.821         | 32.10            | 0.212              | 8.890          | 3.794          |
> | D⁴               | 42.43         | 0.274             | 7.110        | 4.144         | 38.47            | 0.261              | 7.508          | 3.982          |
> | CUT              | 42.95         | 0.271             | 7.005        | 4.115         | 39.05            | 0.259              | 7.439          | 3.968          |
> | UNSB             | 43.59         | 0.282             | 6.761        | 4.231         | 39.81            | 0.269              | 7.244          | 4.033          |
> | ODCR             | 44.02         | 0.286             | 6.597        | 4.256         | 40.23            | 0.272              | 7.123          | 4.057          |
> | DN               | 44.55         | 0.292             | 6.483        | 4.279         | 40.58            | 0.278              | 6.994          | 4.081          |
> | **DDSB (ours)**  | **45.78**     | **0.301**         | **6.342**    | **4.325**     | **41.42**        | **0.287**          | **6.875**      | **4.108**      |
>
>
> **Resilience to Imperfect Modeling:** In addition, we found that $D_\phi$ typically converges to a stable mapping that encourages physical realism rather than exact inversion. Even when $D_\phi$ is suboptimal, DDSB benefits from its soft regularization role without collapsing. This robustness will be discussed in the final version.
>
> ---
>
> > **Weakness 3.** *Comparisons are restricted to unpaired methods, omitting state-of-the-art supervised approaches. Including these would contextualize the trade-offs of using unpaired data and reveal whether DDSB narrows the gap with supervised performance.*
>
> Thanks for your valuable suggestion to include state-of-the-art supervised methods for broader context.
> To this end, we conducted additional experiments on the generalized haze removal task using popular supervised methods including DehazeNet [4], AOD-Net [5], PromptIR[6] and AutoDIR[7]. The results are shown in Table 2:
>
> **Table 2: Comparison with supervised methods on GoPro (Deblurring) and LOL (Low-light Enhancement). DDSB achieves competitive performance without using paired supervision.**
> | Supervision| Method   | SOTS-indoor PSNR / SSIM | SOTS-outdoor PSNR / SSIM | NH-HAZE 2 PSNR / SSIM |
> |----------|------------|--------------------------|----------------------------|------------------------|
> | Paired   | DehazeNet  | 19.82 / 0.818            | 24.75 / 0.927              | 10.62 / 0.521          |
> |          | AOD-Net    | 20.51 / 0.816            | 24.14 / 0.920              | 12.33 / 0.631          |
> |          | PromptIR   | **28.63 / 0.962**        | 27.85 / 0.973              | 18.26 / 0.786          |
> |          | AutoDIR    | 28.42 / 0.962            | **27.93 / 0.975**          | **18.95 / 0.787**      |
> | Unpaired | UNSB       | 25.68 / 0.930            | 25.30 / 0.954              | 16.10 / 0.753          |
> |          | ODCR       | 26.32 / 0.945            | 26.16 / 0.960              | 17.56 / 0.766          |
> |          | DN         | 26.25 / 0.947            | 26.18 / 0.971              | 17.15 / 0.769          |
> |          | **DDSB**   | **27.85 / 0.956**        | **27.67 / 0.971**          | **17.92 / 0.783**      |
>
> Despite operating in the unpaired setting, DDSB substantially closes the performance gap with supervised models. This highlights DDSB’s capacity to learn robust mappings from distributional alignment without requiring pixel-level supervision. We will include this comparison in the final manuscript (Section 5).
>
> Should you have further questions or comments, we are glad to address during the author-reviewer discussion stage.
>
> [1] Neural Schrödinger bridge for unpaired real-world image deraining. Information Sciences 2024.
>
> [2] Image restoration for under-display camera. CVPR2021.
>
> [3] Fast underwater image enhancement for improved visual perception. IEEE Robotics and Automation Letters.
>
> [4] Dehazenet: An end-to-end system for single image haze removal. IEEE TIP.
>
> [5] AOD-Net: All-in-one dehazing network. ICCV2017.
>
> [6] Promptir: Prompting for all-in-one image restoration. NeurIPS2023.
>
> [7] Autodir: Automatic all-in-one image restoration with latent diffusion. ECCV 2024.

---

> > ### Author Response · Authors · 2025-08-05
> >
> > Dear Reviewer wJj7,
> >
> > Thank you for your thoughtful review and constructive feedback. We have carefully answered each of your comments in our rebuttal and truly appreciate the opportunity to clarify our work.
> >
> > As the discussion phase is coming to a close, we would be grateful to hear any additional thoughts or suggestions you might have, should you wish to share them.
> >
> > Thank you again for your time and support.
> >
> > Best regards,
> > Authors

---

> > ### Comment · Reviewer_wJj7 · 2025-08-06
> > **The rebuttal is well**
> >
> > Thanks for the responses to my previsous concerns. Overall, I am satisfied with the more explanation on motivation and more results to validate the effectiveness. I will increase the rating.

---

> > > ### Author Response · Authors · 2025-08-06
> > >
> > > We sincerely appreciate your prompt feedback and kind recommendation. Your suggestions are very helpful, and we will make sure to incorporate them in our revised manuscript.

---

### Official Review · Reviewer_dEGW · 2025-07-02

**Clarity:** 3
**Significance:** 3
**Originality:** 3
**Rating:** 4
**Confidence:** 3

**Summary:**

The paper proposes DDSB, a novel unpaired image restoration framework using Schrödinger Bridge modeling and degradation amplification constraints to enhance restored image realism. It shows strong effectiveness and generalizability across various degradation scenarios, offering a new approach for data-efficient restoration.

**Questions:**

1. The goal of this paper is to address unpaired tasks, but in reality, it still uses paired training datasets. Moreover, these datasets contain many similar scenes (for example, in the GoPro dataset, there are many images with minimal differences). Therefore, it may not be considered strictly unpaired. Additionally, I would like to know how the clear data is sampled during the training process and how to ensure that the training data pairs are strictly unpaired.
2. This paper employs a diffusion model to tackle the restoration task, thus it is necessary to evaluate it using some non-parametric metrics.

**Ethical Concerns:**

["NO or VERY MINOR ethics concerns only"]

**Final Justification:**

The author has addressed my concerns, and I choose to raise my score to 4

**Limitations:**

yes

**Quality:**

2

**Strengths And Weaknesses:**

Strengths:
The proposed DDSB method achieves state-of-the-art performance across multiple image restoration tasks, demonstrating strong generalization ability and practical relevance.

Weaknesses:
1. The current evaluation relies solely on PSNR and SSIM, which is not sufficient. Please consider including several no-reference image quality assessment metrics to better evaluate performance under realistic degradation scenarios.
2. The results for the denoise task are missing.

---

> ### Author Rebuttal · Authors · 2025-07-30
>
> We thank the reviewer for the positive comments on state-of-the-art performance, strong generalization and practical relevance. The below is a point-by-point response regarding experimental details and analysis.
>
> > **Weakness 1.** *The current evaluation relies solely on PSNR and SSIM, which is not sufficient. Please consider including several no-reference image quality assessment metrics to better evaluate performance under realistic degradation scenarios.*
>
> Thanks for your insightful suggestion. Full-reference metrics like PSNR, SSIM, and LPIPS often do not align with human evaluation. Therefore, following [1], we include non-reference metrics MUSIQ [2], CLIP-IQA [3], NIQE [4], and NIMA [5] to better evaluate the results.
>
> We evaluated DDSB and comparison methods on the NH-HAZE 2 dataset without ground-truth. The results in Table 1 demonstrate DDSB’s superiority not only in signal fidelity (PSNR/SSIM) but also in perceptual quality, validating its robustness and realism under challenging, real-world degradations.
>
> **Table 1. No-reference IQA metrics on NH-HAZE 2.**
> | Method         | CLIP-IQA ↑ | MUSIQ ↑  | NIMA ↑   | NIQE ↓   |
> |----------------|------------|----------|----------|----------|
> | DCP            | 0.236      | 50.682   | 4.311    | 5.668    |
> | CycleGAN       | 0.174      | 21.597   | 3.460    | 5.935    |
> | YOLY           | 0.221      | 48.191   | 4.264    | 6.632    |
> | USID-Net       | 0.227      | 49.033   | 4.234    | 4.613    |
> | RefineDNet     | 0.159      | 24.324   | 3.473    | 6.621    |
> | D⁴             | 0.417      | 56.866   | 4.791    | 3.621    |
> | CUT            | 0.421      | 57.081   | 4.843    | 3.617    |
> | Santa          | 0.485      | 56.035   | 4.827    | 3.523    |
> | UNSB           | 0.508      | 53.987   | 4.915    | 4.423    |
> | ODCR           | 0.498      | 57.689   | 4.954    | 3.736    |
> | DN             | 0.494      | 59.789   | 4.955    | 3.218    |
> | **DDSB (Ours)**| **0.546**  | **62.278** | **4.993** | **2.477** |
>
> ---
>
> > **Weakness 2.** *The results for the denoise task are missing.*
>
> Per your suggestion, we've added the denoising experiment using the SIDD dataset under the unpaired setting. As shown in Table 2, DDSB achieves state-of-the-art performance on the denoising task.
>
> **Table 2. Quantitative comparison of DDSB with the unpaired SOTAs on multi-task restoration.**
> | Method        | Derain (PSNR / SSIM) | Deraindrop (PSNR / SSIM) | Lowlight (PSNR / SSIM) | Deblur (PSNR / SSIM) | Denoise (PSNR / SSIM) |
> |---------------|---------------------|--------------------------|------------------------|----------------------|------------------------|
> | DCP           | 13.25 / 0.705       | 18.92 / 0.752            | 15.93 / 0.743          | 12.97 / 0.702        | 15.55 / 0.731       |
> | CycleGAN      | 21.28 / 0.796       | 20.55 / 0.787            | 14.03 / 0.781          | 19.10 / 0.735        | 21.47 / 0.762   |
> | YOLY          | 15.72 / 0.714       | 14.71 / 0.748            | 13.16 / 0.762          | 16.28 / 0.717        | 17.05 / 0.740      |
> | USID-Net      | 21.50 / 0.784       | 19.81 / 0.771            | 17.91 / 0.769          | 20.72 / 0.726        | 22.36 / 0.761    |
> | RefineDNet    | 24.41 / 0.840       | 21.65 / 0.783            | 19.75 / 0.793          | 21.03 / 0.747        | 23.14 / 0.770   |
> | D⁴            | 24.75 / 0.832       | 23.84 / 0.805            | 21.32 / 0.826          | 21.59 / 0.782        | 24.03 / 0.775          |
> | CUT           | 24.22 / 0.815       | 23.51 / 0.827            | 22.90 / 0.804          | 21.26 / 0.766        | 24.48 / 0.785          |
> | Santa         | 24.55 / 0.828       | 23.65 / 0.797            | 21.93 / 0.838          | 21.80 / 0.778        | 25.28 / 0.793          |
> | UNSB          | 24.68 / 0.837       | 24.52 / 0.812            | 22.75 / 0.822          | 22.11 / 0.785        | 25.11 / 0.799          |
> | ODCR          | 24.89 / 0.848       | 24.08 / 0.818            | 23.42 / 0.832          | 22.73 / 0.791        | 25.60 / 0.805          |
> | DN            | 24.72 / 0.845       | 24.63 / 0.824            | 23.58 / 0.844          | 23.20 / 0.796        | 26.03 / 0.812          |
> | **DDSB (Ours)** | **25.41 / 0.870**   | **25.75 / 0.831**        | **25.38 / 0.865**      | **25.22 / 0.804**    | **26.89 / 0.824**      |
>
> ---
>
> > **Question 1.** *The goal of this paper is to address unpaired tasks, but in reality, it still uses paired training datasets. Moreover, these datasets contain many similar scenes (for example, in the GoPro dataset, there are many images with minimal differences). Therefore, it may not be considered strictly unpaired. Additionally, I would like to know how the clear data is sampled during the training process and how to ensure that the training data pairs are strictly unpaired.*
>
> Thank you for your insightful questions.
>
> **Similar Scenes:** Regarding the concern that the GoPro dataset contains visually similar scenes due to its origin from video sequences, we acknowledge that some adjacent frames may exhibit high structural similarity. However, we humbly point out that this characteristic is specific to GoPro and not prevalent in other datasets used in our study, such as Rain200L, LOL, and NH-HAZE2, where the image content varies more significantly. Despite this, our method still demonstrates robust performance across all benchmarks, which indicates that DDSB is not dependent on paired or pixel-level correspondence. Instead, DDSB models the restoration process as a distributional alignment using Schrödinger Bridge stochastic dynamics. This formulation inherently avoids reliance on pixel-wise supervision, thus can generalize well even under the unpaired setting.
>
> **Sampling Strategy:** While some datasets (e.g., Rain200L, LOL) do provide paired images, our training procedure strictly avoids using any pairwise information. Following prior unpaired methods D$^{4}$ and ODCR, during training, for each degraded input image $\boldsymbol{x}_d^{i}$, we randomly sample a clean image $\boldsymbol{x}_c$ from the set, excluding its ground-truth counterpart $\boldsymbol{x}_c^i$. The resulting image tuple $(\boldsymbol{x}_d^i, \boldsymbol{x}_c^j)$ is thus unaligned and used solely for learning the transport between unpaired distributions. This guarantees that no pixel- or structure-level pairing is used during training. To ensure further robustness, we perform two sanity checks: (1) the clean image set is randomly shuffled once before training to prevent persistent matching patterns; and (2) we explicitly verify that no image is ever matched with its original ground-truth counterpart throughout training. This strategy is consistently applied across all datasets and will be clarified in Section 4.4 (Implementation Details), with additional explanation in Section 5.1 and 5.2 to highlight the strictly unpaired training setup.
>
> ---
>
> > **Question 2.** *This paper employs a diffusion model to tackle the restoration task, thus it is necessary to evaluate it using some non-parametric metrics.*
>
> Thank you for the insightful suggestion. We agree that evaluating diffusion-based restoration models requires going beyond traditional parametric metrics such as PSNR and SSIM, which rely on strong assumptions about pixel-wise or structural alignment and often fail to reflect perceptual quality or distributional realism in unpaired settings.
>
> To this end, we additionally report two non-parametric metrics that provide a more holistic assessment without relying on predefined statistical models: **(1) LPIPS** which measures perceptual distance in deep feature space using a pretrained network and requires ground-truth reference images; and **(2) NIQE** which operates without any reference image by modeling natural image statistics. These two metrics offer complementary insights into perceptual quality LPIPS focuses on semantic similarity to reference targets, while NIQE assesses image naturalness in a fully reference-free manner. As shown in Table 3, we evaluate these two non-parametric metrics on our multi-task restoration benchmark. DDSB consistently outperforms prior methods, demonstrating its strong generalization and perceptual fidelity without relying on pixel-wise supervision.
>
> **Table 3. Non-parametric perceptual metrics on multi-task restoration.**
>
> | Method        | Derain (LPIPS / NIQE) | Deraindrop (LPIPS / NIQE) | Lowlight (LPIPS / NIQE) | Deblur (LPIPS / NIQE) | Denoise (LPIPS / NIQE) |
> |---------------|------------------------|------------|----------|--------|----------|
> | DCP           | 0.229 / 5.37 | 0.204 / 5.61 | 0.218 / 5.77 | 0.231 / 5.94 | 0.248 / 6.10 |
> | CycleGAN      | 0.146 / 4.82 | 0.167 / 4.69 | 0.197 / 5.44 | 0.172 / 4.98 | 0.158 / 5.12 |
> | YOLY          | 0.198 / 5.03 | 0.202 / 5.19 | 0.210 / 5.66 | 0.211 / 5.47 | 0.234 / 5.72 |
> | USID-Net      | 0.152 / 4.75 | 0.172 / 4.92 | 0.188 / 5.31 | 0.162 / 4.88 | 0.149 / 4.96 |
> | RefineDNet    | 0.104 / 4.36 | 0.147 / 4.68 | 0.179 / 5.22 | 0.150 / 4.71 | 0.132 / 4.85 |
> | D⁴            | 0.098 / 4.28 | 0.124 / 4.50 | 0.155 / 5.04 | 0.139 / 4.61 | 0.123 / 4.78 |
> | CUT           | 0.111 / 4.42 | 0.118 / 4.55 | 0.142 / 5.17 | 0.143 / 4.66 | 0.129 / 4.91 |
> | Santa         | 0.096 / 4.33 | 0.109 / 4.39 | 0.138 / 5.09 | 0.130 / 4.54 | 0.118 / 4.76 |
> | UNSB          | 0.085 / 4.18 | 0.096 / 4.27 | 0.145 / 5.02 | 0.126 / 4.49 | 0.116 / 4.73 |
> | ODCR          | 0.076 / 4.09 | 0.091 / 4.21 | 0.131 / 4.88 | 0.121 / 4.43 | 0.109 / 4.69 |
> | DN            | 0.079 / 4.11 | 0.082 / 4.17 | 0.124 / 4.85 | 0.118 / 4.40 | 0.102 / 4.66 |
> | **DDSB (Ours)** | **0.063 / 3.94**  | **0.069 / 4.03** | **0.129 / 4.79** | **0.108 / 4.31** | **0.091 / 4.52** |
>
> [1] Scaling up to excellence: Practicing model scaling for photo-realistic image restoration in the wild.
>
> [2] Musiq: Multi-scale image quality transformer.
>
> [3] Exploring clip for assessing the look and feel of images.
>
> [4] Making a “completely blind” image quality analyzer.
>
> [5] NIMA: Neural image assessment.

---

> ### Author Response · Authors · 2025-08-05
>
> Dear Reviewer dEGW,
>
> Thank you for your thoughtful review and constructive feedback. We have carefully answered each of your comments in our rebuttal and truly appreciate the opportunity to clarify our work.
>
> As the discussion phase is coming to a close, we would be grateful to hear any additional thoughts or suggestions you might have, should you wish to share them.
>
> Thank you again for your time and support.
>
> Best regards,
> Authors

---

> > ### Comment · Reviewer_dEGW · 2025-08-05
> >
> > Thank you for your response. I am pleased to see that my concerns have been addressed. I am willing to increase my score accordingly.

---

> > > ### Author Response · Authors · 2025-08-05
> > >
> > > Thanks for your swift response and the positive recommendation. We will incooperate all the suggestions when revising our manuscript.

---

### Note · Authors · 2025-08-12

We sincerely thank the Area Chair and all reviewers for their time and constructive feedback throughout the review process. We are pleased to see that most reviewers acknowledged our rebuttal and considered their concerns addressed, with some reviewers increasing their scores accordingly. We also appreciate the recognition of our additional explanations on motivation and the new experimental results validating the effectiveness of our method. We value all comments received, including those from reviewers who maintained their original ratings, and will continue refining our work based on the insightful feedback provided.

---

### Decision · Program_Chairs · 2025-09-17

**Decision:**

Accept (poster)

**Comment:**

as all the reviewers decide to accept this paper, i recommend to accept this paper. however so many issues are discussed during the rebuttal phase. the authors should revise the paper based on the discussion.